# Doubly Robust Proximal Causal Learning for Continuous Treatments

**Yong Wu**[1,3,4,5] **Yanwei Fu**[2] **Shouyan Wang**[1,3,4,5,6,7] **Xinwei Sun**[2]*

[1]Institute of Science and Technology for Brain-Inspired Intelligence, Fudan University
[2]School of Data Science, Fudan University   [3]Zhangjiang Fudan International Innovation Center
[4]Key Laboratory of Computational Neuroscience and Brain-Inspired Intelligence (Fudan University)
[5]MOE Frontiers Center for Brain Science, Fudan University
[6]Shanghai Engineering Research Center of AI & Robotics, Fudan University
[7]Engineering Research Center of AI & Robotics, Ministry of Education, Fudan University

## Abstract

Proximal causal learning is a powerful framework for identifying the causal effect under the existence of unmeasured confounders. Within this framework, the doubly robust (DR) estimator was derived and has shown its effectiveness in estimation, especially when the model assumption is violated. However, the current form of the DR estimator is restricted to binary treatments, while the treatments can be continuous in many real-world applications. The primary obstacle to continuous treatments resides in the delta function present in the original DR estimator, making it infeasible in causal effect estimation and introducing a heavy computational burden in nuisance function estimation. To address these challenges, we propose a kernel-based DR estimator that can well handle continuous treatments for proximal causal learning. Equipped with its smoothness, we show that its oracle form is a consistent approximation of the influence function. Further, we propose a new approach to efficiently solve the nuisance functions. We then provide a comprehensive convergence analysis in terms of the mean square error. We demonstrate the utility of our estimator on synthetic datasets and real-world applications[1].

## 1 Introduction

The causal effect estimation is a significant issue in many fields such as social sciences (Hedström & Ylikoski, 2010), economics (Varian, 2016), and medicine (Yazdani & Boerwinkle, 2015). A critical challenge in causal inference is non-compliance to randomness due to the presence of unobserved confounders, which can induce biases in the estimation.

One approach to address this challenge is the proximal causal learning (PCL) framework (Miao et al., 2018a; Tchetgen et al., 2020; Cui et al., 2023), which offers an opportunity to learn about causal effects where ignorability condition fails. This framework employs two proxies - a treatment-inducing proxy and an outcome-inducing proxy - to identify the causal effect by estimating the bridge/nuisance functions. Particularly, Cui et al. (2023) derived the doubly robust estimator within the PCL framework, which combines the estimator obtained from the treatment bridge function and the estimator obtained from the outcome bridge function. The doubly robust estimator has been widely used in causal effect estimation (Bang & Robins, 2005), as it is able to tolerate violations of model assumptions of bridge functions.

However, current doubly robust estimators (Cui et al., 2023) within the proximal causal framework mainly focus on binary treatments, whereas the treatments can be continuous in many real-world scenarios, including social science, biology, and economics. For example, in therapy studies, we are not only interested in estimating the effect of receiving the drug but also the effectiveness of the drug dose. Another example comes from the data (Donohue III & Levitt, 2001) that focused on policy-making, where one wishes to estimate the effect of legalized abortion on the crime rate.

---

*Corresponding author
[1] Code is available at `https://github.com/yezichu/PCL_Continuous_Treatment`.

Previous work on causal effect for continuous treatments has focused primarily on the unconfoundedness assumption (Kallus & Zhou, 2018; Colangelo & Lee, 2020). However, extending them within the proximal causal farmework encounters several key challenges. Firstly, the Proximal Inverse Probability Weighting (PIPW) part in the original doubly robust (DR) estimator relies on a delta function centered around the treatment value being analyzed, rendering it impractical for empirically estimating causal effects with continuous treatments. Secondly, deriving the influence function will involve dealing with the Gateaux derivative of bridge functions, which is particularly intricate due to its implicit nature. Lastly, the existing estimation process of bridge functions requires running an optimization for each new treatment, rendering it computationally inefficient for practical applications. In light of these formidable challenges, our contribution lies in addressing the open question of deriving the DR estimator for continuous treatments within the proximal causal framework.

To address these challenges, we propose a kernel-based method that can well handle continuous treatments for PCL. Specifically, we incorporate the kernel function into the PIPW estimator, as a smooth approximation to causal effect. We then derive the DR estimator and show its consistency for a broad family of kernel functions. Equipped with smoothness, we show that such a DR estimator coincides with the influence function. To overcome the computational issue in nuisance function estimation, we propose to estimate the propensity score and incorporate it into a min-max optimization problem, which is sufficient to estimate the nuisance functions for all treatments. We show that our estimator enjoys the $O(n^{-4/5})$ convergence rate in mean squared error (MSE). We demonstrate the utility and efficiency on synthetic data and the policy-making (Donohue III & Levitt, 2001).

**Contributions.** To summarize, our contributions are:

1. We propose a kernel-based DR estimator that is provable to be consistent for continuous treatments effect within the proximal causal framework.

2. We efficiently solve bridge functions for all treatments with only a single optimization.

3. We present the convergence analysis of our estimator in terms of MSE.

4. We demonstrated the utility of our estimator on two synthetic data and real data.

## 2 BACKGROUND

**Proximal Causal Learning.** The proximal causal learning (PCL) can be dated back to Kuroki & Pearl (2014), which established the identification of causal effects in the presence of unobserved confounders under linear models. Then Miao et al. (2018a;b) and its extensions (Shi et al., 2020; Tchetgen et al., 2020) proposed to leverage two proxy variables for causal identification by estimating the outcome bridge function. Building upon this foundation, Cui et al. (2023) introduced a treatment bridge function and incorporated it into the Proximal Inverse Probability Weighting (PIPW) estimator. Besides, under binary treatments, they derived the Proximal Doubly Robust (PDR) estimator via influence functions. However, continuous treatments pose a challenge as the treatment effect is not pathwise differentiable with respect to them, preventing the derivation of a DR estimator. In this paper, we employ the kernel method that is provable to be consistent in treatment effect estimation. We further show that the kernel-based DR estimator can be derived from influence functions.

**Causal inference for Continuous Treatments.** The most common approaches for estimating continuous treatment effects are regression-based models (Imbens, 2004; Hill, 2011), generalized propensity score-based models (Imbens, 2000; Hirano & Imbens, 2004; Imai & Van Dyk, 2004), and entropy balance-based methods (Hainmueller, 2012; Imai & Ratkovic, 2014; Tübbicke, 2021). Furthermore, Kennedy et al. (2017); Kallus & Zhou (2018) and Colangelo & Lee (2020) extended the DR estimation to continuous treatments by combining regression-based models and the generalized propensity score-based models. However, it remains open to derive the DR estimator for continuous treatments within the proximal causal framework. In this paper, we fill in this blank with a new kernel-based DR estimator that is provable to derive from influence function.

**Nuisance Parameters Estimation.** In proximal causal learning, one should estimate nuisance parameters to obtain the causal effect. Many methods have been proposed for this goal (Tchetgen et al., 2020; Singh, 2020; Xu et al., 2021; Kompa et al., 2022), but they primarily focus on the estimation of the outcome bridge function. Recently, Kallus et al. (2021); Ghassami et al. (2022) have provided non-parametric estimates of treatment bridge function, but they are restricted to binary treatments. When it comes to continuous treatments, existing methods can be computationally inefficient since

it has to resolve an optimization problem for each treatment. In this paper, we propose a new method that can efficiently solve bridge functions for all treatments with only a single optimization.

## 3 PROXIMAL CAUSAL INFERENCE

**Problem setup.** We consider estimating the Average Causal Effect (ACE) of a continuous treatment $A$ on an outcome $Y$: $\mathbb{E}[Y(a)]$, where $Y(a)$ for any $a \in \mathrm{supp}(A)$ denotes the potential outcome when the treatment $A = a$ is received. We respectively denote $X$ and $U$ as observed covariates and unobserved confounders. To estimate $\mathbb{E}[Y(a)]$, we assume the following consistency assumptions that are widely adopted in causal inference (Peters et al., 2017):

**Assumption 3.1** (Consistency and Positivity). *We assume (i) $Y(A) = Y$ almost surely (a.s.); and (ii) $0 < p(A = a|U = u, X = x) < 1$ a.s.*

**Assumption 3.2** (Latent ignorability). *We assume $Y(a) \perp A|U, X$.*

Assump. 3.2 means that the strong ignorability condition may fail due to the presence of unobserved confounder $U$. To account for such a confounding bias, the proximal causal learning incorporates a treatment-inducing proxy $Z$ and an outcome-inducing proxy $W$. As illustrated in Fig. 1, these proxies should satisfy the following conditional independence:

**Assumption 3.3** (Conditional Independence of Proxies). *The treatment-inducing proxy $Z$ and the outcome-inducing proxy $W$ satisfy the following conditional independence:* **(i)** $Y \perp Z \mid A, U, X$; *and* **(ii)** $W \perp (A, Z) \mid U, X$.

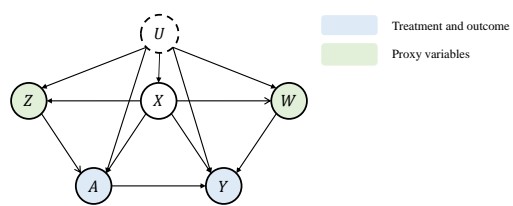

Figure 1: Illustration of causal DAG in proximal causal learning, where $Z, W$ are proxy variables.

Equipped with such conditional independence, previous work by Miao et al. (2018a); Cui et al. (2023) demonstrated that we can express the causal effect, denoted as $\beta(a)$, as follows:

$$\mathbb{E}[Y(a)] = \mathbb{E}[h_0(a, W, X)] = \mathbb{E}[\mathbb{I}(A = a)q_0(a, Z, X)Y], \tag{1}$$

where $h_0$ and $q_0$ are two nuisance/bridge functions such that the following equations hold:

$$\mathcal{R}_h(h_0; y) := \mathbb{E}[Y - h_0(A, W, X)|A, Z, X] = 0, \tag{2}$$
$$\mathcal{R}_q(q_0; p) := \mathbb{E}[q_0(A, Z, X) - 1/p(A|W, X)|A, W, X] = 0. \tag{3}$$

To ensure the existence and uniqueness of solutions to the above equations, we additionally assume that (Miao et al., 2018a; Tchetgen et al., 2020; Cui et al., 2023):

**Assumption 3.4.** *Let $\nu$ denote any square-integrable function. For any $(a, x)$, we have*

1. *(Completeness for outcome bridge functions). We assume that $\mathbb{E}[\nu(U)|W, a, x] = 0$ and $\mathbb{E}[\nu(Z)|W, a, x] = 0$ iff $\nu(U) = 0$ almost surely.*
2. *(Completeness for treatment bridge functions). We assume that $\mathbb{E}[\nu(U)|Z, a, x] = 0$ and $\mathbb{E}[\nu(W)|Z, a, x] = 0$ iff $\nu(U) = 0$ almost surely.*

Under assump. 3.4, we can solve $h_0$ and $q_0$ via several optimization approaches derived from conditional moment equations, including two-stage penalized regression (Singh, 2020; Mastouri et al., 2021; Xu et al., 2021), maximum moment restriction (Zhang et al., 2020; Muandet et al., 2020a), and minimax optimization (Dikkala et al., 2020; Muandet et al., 2020b; Kallus et al., 2021). With solved $h_0, q_0$, we can estimate $\mathbb{E}[Y(a)]$ via:

$$\mathbb{E}_n[Y(a)] = \frac{1}{n}\sum_{i=1}^n h_0(a_i, w_i, x_i), \ \ \text{or} \ \ \mathbb{E}_n[Y(a)] = \frac{1}{n}\sum_{i=1}^n \mathbb{I}(a_i = a)q_0(a, z_i, x_i)y_i.$$

Furthermore, Cui et al. (2023) proposes a doubly robust estimator to improve robustness against misspecification of bridge functions.

$$\mathbb{E}[Y(a)] = \mathbb{E}[\mathbb{I}(A = a)q_0(a, Z, X)(Y - h_0(a, W, X)) + h_0(a, W, X)], \tag{4}$$

$$\approx \frac{1}{n}\sum_{i=1}^n (\mathbb{I}(A = a)q_0(a, z_i, x_i)(y_i - h_0(a, w_i, x_i)) + h_0(a, w_i, x_i)). \tag{5}$$

Although this proximal learning method can efficiently estimate $\mathbb{E}[Y(a)]$ for binary treatments, it suffers from several problems when it comes to continuous treatments. First, for any $a \in \text{supp}(A)$, it almost surely holds that there does not exist any sample $i$ that satisfies $a_i = a$ for $i = 1, ..., n$, making Eq. 5 infeasible. Besides, it is challenging to derive the influence function for continuous treatments as it involves the derivative computation for implicit functions $h_0$ and $q_0$. Lastly, to estimate $q_0$, previous methods suffered from a large computational cost since they had to re-run the optimization algorithm for each new treatment, making it inapplicable in real-world applications.

To resolve these problems for continuous treatment, we first introduce a kernel-based method in Sec. 4, which can estimate $\mathbb{E}[Y(a)]$ in a feasible way. Then in Sec. 5, we introduce a new optimization algorithm that can estimate $h_0, q_0$ for all treatments with a single optimization algorithm. Finally, we present the theoretical results in Sec. 6.

## 4    PROXIMAL CONTINUOUS ESTIMATION

In this section, we introduce a kernel-based doubly robust estimator for $\beta(a) := \mathbb{E}[Y(a)]$ with continuous treatments. We first present the estimator form in Sec. 4.1, followed by Sec. 4.2 to show that such an estimator can well approximate the influence function for $\beta(a)$.

### 4.1    KERNEL-BASED PROXIMAL ESTIMATION

As mentioned above, the main challenge for continuous treatments lies in the estimation infeasibility caused by the indicator function in the proximal inverse probability weighted estimator (PIPW) with $q_0$: $\hat{\beta}(a) = \frac{1}{n} \sum_{i=1}^{n} \mathbb{I}(a_i = a) q_0(a, z_i, x_i) y_i$. To resolve this problem, we note that the indicator function can be viewed as a Dirac delta function $\delta_a(a_i)$. The average of this Dirac delta function over $n$ samples $\frac{1}{n} \sum_{i=1}^{n} \delta_a(a_i)$ approximates to the marginal probability $\mathbb{P}(a)$ (Doucet et al., 2009), which equals to 0 when $A$ is continuous.

To address this problem, we integrate the kernel function $K(A - a)$ that can alleviate the unit concentration of the Dirac delta function. We can then rewrite the PIPW estimator as follows, dubbed as **P**roximal **K**ernel **I**nverse **P**robability **W**eighted (PKIPW) estimator:

$$\hat{\beta}(a) = \frac{1}{n} \sum_{i=1}^{n} K_{h_{\text{bw}}}(a_i - a) q_0(a, z_i, x_i) y_i, \tag{6}$$

where $h_{\text{bw}} > 0$ is the bandwidth such that $K_{h_{\text{bw}}}(a_i - a) = \frac{1}{h_{\text{bw}}} K\left(\frac{a_i - a}{h_{\text{bw}}}\right)$. The kernel function $K_{h_{\text{bw}}}(A - a)$ that has been widely adopted in density estimation, assigns a non-zero weight to each sample, thus making it feasible to estimate $\beta(a)$. To demonstrate its validity, we next show that it can approximate $\beta(a)$ well. This result requires that the kernel function $K$ is bounded differentiable, as formally stated in the following.

**Assumption 4.1.** *The second-order symmetric kernel function $K\left(\cdot\right)$ is bounded differentiable, i.e., $\int k(u)\mathrm{d}u = 1, \int u k(u)\mathrm{d}u = 0, \kappa_2(K) = \int u^2 k(u)\mathrm{d}u < \infty$. We define $\Omega_2^{(i)}(K) = \int (k^{(i)}(u))^2 \mathrm{d}u$.*

Assump. 4.1 adheres to the conventional norms within the domain of nonparametric kernel estimation and maintains its validity across widely adopted kernel functions, including but not limited to the Epanechnikov and Gaussian kernels. Under assump. 4.1, we have the following theorem:

**Theorem 4.2.** *Under assump. 4.1, suppose $\beta(a) = \mathbb{E}[\mathbb{I}(A = a) q_0(a, Z, X) Y]$ is continuous and bounded uniformly respect to $a$, then we have*

$$\mathbb{E}[Y(a)] = \mathbb{E}[\mathbb{I}(A = a) q_0(a, Z, X) Y] = \lim_{h_{\text{bw}} \to 0} \mathbb{E}\left[K_{h_{\text{bw}}}(A - a) q_0(a, Z, X) Y\right],$$

*Remark* 4.3. The kernel function has been widely used in machine learning applications (Kallus & Zhou, 2018; Kallus & Uehara, 2020; Colangelo & Lee, 2020; Klosin, 2021). Different from these works, we are the first to integrate them into the proximal estimation to handle continuous treatments.

*Remark* 4.4. The choice of bandwidth $h_{\text{bw}}$ is a trade-off between bias and variance. When $h_{\text{bw}}$ is small, the kernel estimator has less bias as shown in Thm. 4.2, however, will increase the variance. In Sec. 6, we show that the optimal rate for $h_{\text{bw}}$ is $O(n^{-1/5})$, which leads to the MSE converges at a rate of $O(n^{-4/5})$ for our kernel-based doubly robust estimator.

Similar to Eq. 6, we can therefore derive the **P**roximal **K**ernel **D**oubly **R**obust (PKDR) estimator as:

$$\hat{\beta}(a) = \frac{1}{nh_{\mathrm{bw}}} \sum_{i=1}^{n} K\left(\frac{a_i - a}{h_{\mathrm{bw}}}\right)(y_i - h_0(a, w_i, x_i))\, q_0(a, z_i, x_i) + h_0(a, w_i, x_i). \tag{7}$$

Similar to Thm. 4.2, we can also show that this estimator is unbiased as $h_{\mathrm{bw}} \to 0$. In the subsequent section, we show that this estimator in Eq. 7 can also be derived from the smooth approximation of the influence function of $\beta(a)$.

## 4.2 Influence Function under Continuous Treatments

In this section, we employ the method of Gateaux derivative (Carone et al., 2018; Ichimura & Newey, 2022) to derive the influence function of $\beta(a)$. (For our non-regular parameters, we borrow the terminology "influence function" in estimating a regular parameter. See Hampel Ichimura & Newey (2022), for example.) Specifically, we denote $\mathbb{P}_X$ as the distribution function for any variable $X$, and rewrite $\beta(a)$ as $\beta(a; \mathbb{P}_O^0)$ where $\mathbb{P}_O^0$ denotes the true distribution for $O := (A, Z, W, X, Y)$. Besides, we consider the special submodel $\mathbb{P}_O^{\varepsilon h_{\mathrm{bw}}} = (1 - \varepsilon)\mathbb{P}_O^0 + \varepsilon \mathbb{P}_O^{h_{\mathrm{bw}}}$, where $\mathbb{P}_O^{h_{\mathrm{bw}}}(\cdot)$ maps a point $o$ to a distribution of $O$, i.e., $\mathbb{P}_O^{h_{\mathrm{bw}}}(o)$ for a fixed $o$ denotes the distribution of $O$ that approximates to a point mass at $o$. Different types of $\mathbb{P}_O^{h_{\mathrm{bw}}}(o)$ lead to different forms of Gateaux derivative. In our paper, we choose the distribution $\mathbb{P}_O^{h_{\mathrm{bw}}}(o)$ whose corresponding probability density function (pdf) $p_O^{h_{\mathrm{bw}}}(o) = K_{h_{\mathrm{bw}}}(O - o)\mathbb{I}(p_O^0(o) > h_{\mathrm{bw}})$, which has $\lim_{h_{\mathrm{bw}} \to 0} p_O^{h_{\mathrm{bw}}}(o) = \lim_{h_{\mathrm{bw}} \to 0} K_{h_{\mathrm{bw}}}(O - o)$.

We can then calculate the limit of the Gateaux derivative (Ichimura & Newey, 2022) of the functional $\beta(a; \mathbb{P}_O^{\varepsilon h_{\mathrm{bw}}})$ with respect to a deviation $\mathbb{P}_O^{h_{\mathrm{bw}}} - \mathbb{P}_O^0$. The following theorem shows that our kernel-based doubly robust estimator corresponds to the influence function:

**Theorem 4.5.** *Under a nonparametric model, the limit of the Gateaux derivative is*

$$\lim_{h_{\mathrm{bw}} \to 0} \frac{\partial}{\partial \varepsilon} \beta(a; \mathbb{P}^{\varepsilon h_{\mathrm{bw}}})\Big|_{\varepsilon=0} = (Y - h_0(a, W, X))\, q_0(a, Z, X) \lim_{h_{\mathrm{bw}} \to 0} K_{h_{\mathrm{bw}}}(A - a) + h_0(a, W, X) - \beta(a)$$

*Remark* 4.6. For binary treatments, the DR estimator with the indicator function in Eq. 4 corresponds to the efficient influence function, as derived within the non-parametric framework (Cui et al., 2023). Different from previous works Colangelo & Lee (2020), deriving the influence function within the proximal causal framework is much more challenging as it involves the Gateau derivatives for nuisance functions $h_0, q_0$ that have implicit functional forms. By employing our estimator, even when the unconfoundedness assumption from Colangelo & Lee (2020) is not satisfied, we can still effectively obtain causal effects.

## 5 Nuisance function estimation

In this section, we propose to solve $h_0, q_0$ from integral equations Eq. 2, 3 for continuous treatments. We first introduce the estimation of $q_0$. Previous methods (Kallus et al., 2021; Ghassami et al., 2022) solved $q_0(a, Z, X)$ by running an optimization algorithm for each $a = 0, 1$. However, it is computationally infeasible for continuous treatments. Please see Appx. D.2 for detailed comparison. Instead of running an optimization for each $a$, we would like to estimate $q_0(A, Z, X)$ with a single optimization algorithm. To achieve this goal, we propose a two-stage estimation algorithm. We first estimate the policy function $p(A|w, x)$ and plug into Eq. 3. To efficiently solve $q_0$, we note that it is equivalent to minimize the residual mean squared error denoted as $\mathcal{L}_q(q; p) = \mathbb{E}[(\mathcal{R}_q(q, p))^2]$. According to the lemma shown below, such a mean squared error can be reformulated into a maximization-style optimization, thereby converting into a min-max optimization problem.

**Lemma 5.1.** *Denote $\|f(X)\|_{L_2}^2 := \mathbb{E}[f^2(X)]$. For any parameter $\lambda_m > 0$, we have*

$$\mathcal{L}_q(q; p) = \sup_{m \in \mathcal{M}} \mathbb{E}\left[m(A, W, X)\left(q_0(A, Z, X) - 1/p(A|W, X)\right)\right] - \lambda_m \|m(A, W, X)\|_{L_2}^2,$$

*where $\mathcal{M}$ is the space of continuous functions over $(A, W, X)$.*

We leave the proof in Appx. D. Motivated by Lemma. 5.1, we can solve $q_0$ via the following min-max optimization:

$$\min_{q \in \mathcal{Q}} \max_{m \in \mathcal{M}} \Phi_q^{n, \lambda_m}(q, m; p) := \frac{1}{n} \sum_i \left(q(a_i, z_i, x_i) - \frac{1}{p(a_i|w_i, x_i)}\right) m(a_i, w_i, x_i) - \lambda_m \|m\|_{2,n}^2,$$

$$\tag{8}$$

where $\lambda_m\|m\|_{2,n}^2$ is called stabilizer with $\|m\|_{2,n}^2 := \frac{1}{n}\sum_i m^2(a_i, w_i, x_i)$. We can parameterize $q$ and $m$ as reproducing kernel Hilbert space (RKHS) with kernel function to solve the min-max problem. We derive their closed solutions in the Appendix F. Besides, we can also use Generative Adversarial Networks (Goodfellow et al., 2014) to solve this problem.

**Estimating the policy function** $p(A = a|w, x)$**.** To optimize Eq. 8, we should first estimate $p(a|w, x)$. Several methods can be used for this estimation, such as the kernel density estimation and normalizing flows (Chen, 2017; Bishop, 1994; Ambrogioni et al., 2017; Sohn et al., 2015; Rezende & Mohamed, 2015; Dinh et al., 2016). In this paper, we employ the kernel density function (Chen, 2017) that has been shown to be effective in low-dimension scenarios. When the dimension of $(W, X)$ is high, we employ the conditional normalizing flows (CNFs), which have been shown to be universal density approximator (Durkan et al., 2019) and thus can be applied to complex scenarios.

**Nuisance function** $h_0$**.** Since the estimation of $h_0$ does not involve indicator functions, we can apply many off-the-shelf optimization approaches derived from conditional moment equations, such as two-stage penalized regression (Singh, 2020; Mastouri et al., 2021; Xu et al., 2021), maximum moment restriction (Zhang et al., 2020; Muandet et al., 2020a), and minimax optimization (Dikkala et al., 2020; Muandet et al., 2020b). To align well with $q_0$, here we choose to estimate $h_0$ via the following min-max optimization problem that has been derived in Kallus et al. (2021):

$$\min_{h\in\mathcal{H}}\max_{g\in\mathcal{G}}\Phi_h^{n,\lambda_g}(h,g) := \frac{1}{n}\sum_i g(a_i, z_i, x_i)\left(y_i - h(a_i, w_i, x_i)\right) - \lambda_g\|g\|_{2,n}^2, \qquad (9)$$

where $\mathcal{H}$ and $\mathcal{G}$ respectively denote the bridge functional class and the critic functional class.

## 6 THEORETICAL RESULTS

In this section, we provide convergence analysis of Eq. 8, 9 for nuisance functions $h_0, q_0$, as well as for the causal effect $\beta(a)$ with the PKDR estimator in Eq. 7.

We first provide convergence analysis for $q_0$, while the result for $h_0$ is similar and left to the Appx. E. Different from previous works Dikkala et al. (2020); Ghassami et al. (2022), our analysis encounters a significant challenge arising from the estimation error inherent in the propensity score function. By addressing this challenge, our result can effectively account for this estimation error.

Formally speaking, we consider the projected residual mean squared error (RMSE) $\mathbb{E}[\text{proj}_q(\hat{q} - q_0)^2]$, where $\text{proj}_q(\cdot) := \mathbb{E}[\cdot|A, W, X]$. Before presenting our results, we first introduce the assumption regarding the critic functional class in $\mathcal{M}$, which has been similarly made in Dikkala et al. (2020); Ghassami et al. (2022); Qi et al. (2023).

**Assumption 6.1.** (1) (Boundness) $\|\mathcal{Q}\|_\infty < \infty$ and $\hat{p}$ is uniformly bounded; (2) (Symmetric) $\mathcal{M}$ is a symmetric class, i.e, if $m \in \mathcal{M}$, then $-m \in \mathcal{M}$; (3) (Star-shaped) $\mathcal{M}$ is star-shaped class, that is for each function $m$ in the class, $\alpha m$ for any $\alpha \in [0, 1]$ also belongs to the class; (4) (Realizability) $q_0 \in \mathcal{Q}$; (5) (Closedness) $\frac{1}{2\lambda_m}\text{proj}_q(q - q_0) \in \mathcal{M}$.

Under assumption 6.1, we have the following convergence result in terms of $\|\text{proj}_q(\hat{q} - q_0)\|_{L_2}$.

**Theorem 6.2.** *Let $\delta_n^q$ respectively be the upper bound on the Rademacher complexity of $\mathcal{M}$. For any $\eta \in (0, 1)$, define $\delta^q := \delta_n^q + c_0^q\sqrt{\frac{\log(c_1^q/\eta)}{n}}$ for some constants $c_0^q, c_1^q$; then under assump. 6.1, we have with probability $1 - \eta$ that*

$$\left\|\text{proj}_q(\hat{q} - q_0)\right\|_2 = O\left(\delta^q\sqrt{\lambda_m^2 + \lambda_m + 1} + \left\|\frac{1}{p} - \frac{1}{\hat{p}}\right\|_2\right), \ p \ \text{stands for} \ p(a|w, x).$$

*Remark* 6.3. Inspired by Chen & Pouzo (2012); Dikkala et al. (2020); Kallus et al. (2021), we can obtain the same upper bound for the RMSE $\|\hat{q} - q_0\|_2$, up to a measure of ill-posedness denoted as $\tau_q := \sup_{q\in\mathcal{Q}}\|q - q_0\|_2/\|\text{proj}_q(q - q_0)\|_2 < \infty$.

The bound mentioned above comprises two components. The first part pertains to the estimation of $q$, while the second part concerns the estimation of $1/p$. The first part is mainly occupied by the Rademacher complexity $\delta_n^q$, which can attain $O(n^{-1/4})$ if we parameterize $\mathcal{M}$ as bounded metric entropy such as Holder balls, Sobolev balls, and RKHSs. For the second part, we can also

achieve $O(n^{-1/4})$ for $\|1/p - 1/\hat{p}\|_2$ under some conditions (Chernozhukov et al., 2022; Klosin, 2021; Colangelo & Lee, 2020).

Now we are ready to present the convergence result for $\beta(a)$ within the proximal causal framework.

**Theorem 6.4.** *Under assump. 3.1-3.4 and 4.1, suppose $\|\hat{h} - h\|_2 = o(1)$, $\|\hat{q} - q\|_2 = o(1)$ and $\|\hat{h} - h\|_2\|\hat{q} - q\|_2 = o((nh_{\mathrm{bw}})^{-1/2}), nh_{\mathrm{bw}}^5 = O(1), nh_{\mathrm{bw}} \to \infty, h_0(a, w, x), p(a, z|w, x)$ and $p(a, w|z, x)$ are twice continuously differentiable wrt $a$ as well as $h_0, q_0, \hat{h}, \hat{q}$ are uniformly bounded. Then for any $a$, we have the following for the bias and variance of the PKDR estimator given Eq. 7:*

$$\mathrm{Bias}(\hat{\beta}(a)) := \mathbb{E}[\hat{\beta}(a)] - \beta(a) = \frac{h_{\mathrm{bw}}^2}{2}\kappa_2(K)B + o((nh_{\mathrm{bw}})^{-1/2}), \mathrm{Var}[\hat{\beta}(a)] = \frac{\Omega_2(K)}{nh_{\mathrm{bw}}}(V + o(1)),$$

*where $B = \mathbb{E}[q_0(a, Z, X)[2\frac{\partial}{\partial A}h_0(a, W, X)\frac{\partial}{\partial A}p(a, W \mid Z, X) + \frac{\partial^2}{\partial A^2}h_0(a, W, X)]], V = \mathbb{E}[\mathbb{I}(A = a)q_0(a, Z, X)^2(Y - h_0(a, W, X))^2].*

*Remark* 6.5. The smoothness condition can hold for a broad family of distributions and be thus similarly made for kernel-based methods (Kallus & Zhou, 2018; Kallus & Uehara, 2020). According to Thm. 6.2, we have $\|\hat{h} - h_0\|_2 = O(n^{-1/4})$ and $\|\hat{q} - q_0\|_2 = O(n^{-1/4})$, thus can satisfy the consistency condition required as long as $h_{\mathrm{bw}} = o(1)$. Besides, we show in Thm. E.9 in Appx. E.5 that this estimator is $n^{2/5}$-consistent.

From Thm. 6.4, we know that the optimal bandwidth is $h_{\mathrm{bw}} = O(n^{-1/5})$ in terms of MES that converges at the rate of $O(n^{-4/5})$. Note that this rate is slower than the optimal rate $O(n^{-1})$, which is a reasonable sacrifice to handle continuous treatment within the proximal causal framework and agrees with existing studies (Kennedy et al., 2017; Colangelo & Lee, 2020).

## 7 EXPERIMENTS

In this section, we evaluate the effectiveness of our method using two sets of synthetic data — one in a low-dimensional context and the other in a high-dimensional context — as well as the legalized abortion and crime dataset (Donohue III & Levitt, 2001). In Appx. G, we conduct experiments on more benchmark datasets, including time-series forecasting.

**Compared baselines.** We compare our method with the following baselines that use only $h_0$ for estimation, *i.e.*, $\hat{\beta}(a) = \frac{1}{n}\sum_{i=1}^{n} h_0(a_i, w_i, x_i)$: **(i)** Proximal Outcome Regression (**POR**) that solved Eq. 9 for estimation; **ii) PMMR** (Mastouri et al., 2021) that employed the Maximal Moment Restriction (MMR) framework to estimate the bridge function via kernel learning; **iii) KPV** (Mastouri et al., 2021) that used two-stage kernel regression; **iv) DFPV** (Xu et al., 2021) that used deep neural networks to model high-dimensional nonlinear relationships between proxies and outcomes; **v) MINMAX** (Dikkala et al., 2020) that used Generate adversarial networks to solve Eq. 9; **vi) NMMR** (Kompa et al., 2022) that introduced data-adaptive kernel functions derived from neural networks.

For our method, we implement the Inverse probability weighting (IPW) estimator **PKIPW** that uses $q_0$ for estimation via Eq. 6, and the doubly robust estimator **PKDR** that used both the nuisance function $h_0$ and $q_0$ to estimate causal effects through Eq. 7. For simplicity, we only present the result of PKDR that uses POR to estimate $h_0$.

**Implementation Details.** In the PKIPW and PKDR estimators, we choose the second-order Epanechnikov kernel, with bandwidth $h_{\mathrm{bw}} = c\hat{\sigma}_A n^{-1/5}$ with estimated std $\hat{\sigma}_A$ and the hyperparameter $c > 0$. In our paper, we vary $c$ over the range $\{0.5, 1, 1.5, \cdots, 4.0\}$ and report the optimal $c$ in terms of cMSE. To estimate nuisance functions, we parameterize $\mathcal{Q}$ and $\mathcal{M}$ (resp., $\mathcal{H}$ and $\mathcal{G}$) via RKHS for $q_0$ (resp., $h_0$), where we use Gaussian kernels with the bandwidth parameters being initialized using the median distance heuristic. For policy estimation, we employ the KDE in the low-dimensional synthetic dataset and the real-world data, while opting for CNFs in the high-dimensional synthetic dataset. We leave more details about hyperparameters in the Appx. H.

**Evaluation metrics.** We report the causal Mean Squared Error (cMSE) across 100 equally spaced points in the range of $\mathrm{supp}(A)$: $\mathrm{cMSE} := \frac{1}{100}\sum_{i=1}^{100}(\mathbb{E}[Y^{a_i}] - \hat{\mathbb{E}}[Y^{a_i}])^2$. Here, we respectively take $\mathrm{supp}(A) := [-1, 2], [0, 1], [0, 2]$ in low-dimensional synthetic data, high-dimensional synthetic data, and real-world data. The truth $\mathbb{E}[Y^a]$ is derived through Monte Carlo simulations comprising 10,000 replicates of data generation for each $a$.

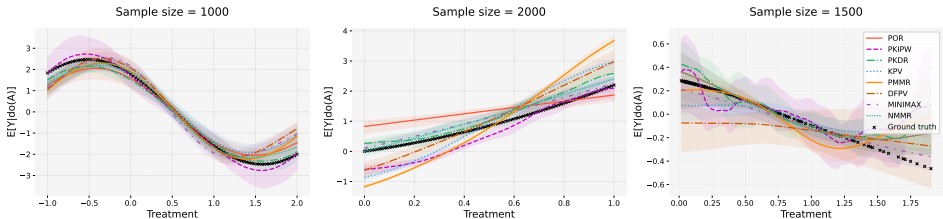

Figure 2: ATE comparison of different methods across various methods on three datasets; Left: ATE comparison using 1000 samples in the first experiment; Middle: ATE comparison using 2000 samples in the second experiment; Right: ATE comparison for the abortion and crime dataset.

## 7.1 SYNTHETIC STUDY

We consider two distinct scenarios. The first scenario demonstrates the effectiveness of the kernel method in the context of the doubly robust estimator under model misspecification, while the second scenario evaluates the utility in high-dimensional settings. For both scenarios, we report the mean cMSE of each method across 20 times.

### 7.1.1 DOUBLY ROBUSTNESS STUDY

**Data generation.** We follow the generative process in Mastouri et al. (2021) and leave details in the Appx. H. Similar to Kang & Schafer (2007); Cui et al. (2023), we consider four scenarios where either or both confounding bridge functions are misspecified by considering a model using a transformation of observed variables:

- **Scenario 1.** We follow Mastouri et al. (2021) to generate data;
- **Scenario 2.** The outcome confounding bridge function is misspecified with $W^* = |W|^{1/2} + 1$;
- **Scenario 3.** The treatment confounding bridge function is misspecified with $Z^* = |Z|^{1/2} + 1$;
- **Scenario 4.** Both confounding bridge functions are mis-specified.

Table 1: cMSE of all methods on two synthetic data and the real-world data.

| Dataset | | Size | PMMR | KPV | DFPV | MINIMAX | NMMR | POR | **PKIPW** | **PKDR** |
|---|---|---|---|---|---|---|---|---|---|---|
| Doubly Robust | Scenario 1 | 500 | $0.16_{\pm0.05}$ | $0.37_{\pm0.26}$ | $0.30_{\pm0.13}$ | $0.20_{\pm0.15}$ | $0.26_{\pm0.11}$ | $0.19_{\pm0.11}$ | $0.11_{\pm0.06}$ | $\mathbf{0.11_{\pm0.06}}$ |
| | | 1000 | $0.14_{\pm0.03}$ | $0.21_{\pm0.09}$ | $0.26_{\pm0.06}$ | $0.10_{\pm0.05}$ | $0.25_{\pm0.09}$ | $0.16_{\pm0.10}$ | $0.11_{\pm0.08}$ | $\mathbf{0.08_{\pm0.04}}$ |
| | Scenario 2 | 500 | $3.32_{\pm0.06}$ | $3.50_{\pm0.16}$ | $1.03_{\pm0.19}$ | $7.48_{\pm2.05}$ | $4.72_{\pm1.38}$ | $3.47_{\pm0.08}$ | $0.16_{\pm0.11}$ | $\mathbf{0.16_{\pm0.09}}$ |
| | | 1000 | $3.32_{\pm0.06}$ | $3.49_{\pm0.19}$ | $0.97_{\pm0.18}$ | $5.27_{\pm0.96}$ | $5.71_{\pm1.10}$ | $3.48_{\pm0.07}$ | $\mathbf{0.20_{\pm0.14}}$ | $0.24_{\pm0.19}$ |
| | Scenario 3 | 500 | $\mathbf{0.15_{\pm0.05}}$ | $0.29_{\pm0.18}$ | $0.28_{\pm0.13}$ | $0.33_{\pm0.26}$ | $0.38_{\pm0.15}$ | $0.20_{\pm0.12}$ | $2.38_{\pm0.60}$ | $0.20_{\pm0.15}$ |
| | | 1000 | $\mathbf{0.14_{\pm0.03}}$ | $0.20_{\pm0.08}$ | $0.30_{\pm0.10}$ | $0.27_{\pm0.13}$ | $0.47_{\pm0.37}$ | $0.22_{\pm0.14}$ | $2.15_{\pm0.90}$ | $0.19_{\pm0.10}$ |
| | Scenario 4 | 500 | $3.32_{\pm0.05}$ | $3.51_{\pm0.21}$ | $\mathbf{1.00_{\pm0.22}}$ | $5.60_{\pm1.03}$ | $6.69_{\pm1.26}$ | $3.46_{\pm0.08}$ | $2.91_{\pm5.58}$ | $3.38_{\pm5.03}$ |
| | | 1000 | $3.29_{\pm0.03}$ | $3.46_{\pm0.14}$ | $\mathbf{1.02_{\pm0.23}}$ | $5.10_{\pm0.90}$ | $5.65_{\pm1.51}$ | $3.44_{\pm0.07}$ | $1.87_{\pm0.91}$ | $3.56_{\pm3.67}$ |
| High Dimension | | 1000 | $0.74_{\pm0.09}$ | $0.34_{\pm0.06}$ | $0.22_{\pm0.11}$ | $0.12_{\pm0.07}$ | $0.14_{\pm0.05}$ | $0.31_{\pm0.07}$ | $0.20_{\pm0.03}$ | $\mathbf{0.08_{\pm0.04}}$ |
| | | 2000 | $0.69_{\pm0.05}$ | $0.36_{\pm0.02}$ | $0.24_{\pm0.09}$ | $0.07_{\pm0.03}$ | $\mathbf{0.05_{\pm0.04}}$ | $0.30_{\pm0.08}$ | $0.19_{\pm0.03}$ | $0.09_{\pm0.04}$ |
| Abort. & Crim | | 1500 | $0.02_{\pm0.00}$ | $0.03_{\pm0.01}$ | $0.07_{\pm0.05}$ | $0.05_{\pm0.00}$ | $0.01_{\pm0.01}$ | $\mathbf{0.01_{\pm0.00}}$ | $0.04_{\pm0.02}$ | $0.02_{\pm0.00}$ |

**Results.** We present the mean and the standard deviation (std) of cMSE over 20 times across four scenarios, as depicted in Fig. 2 and Tab. 1. For each scenario, we consider two sample sizes, 500 and 1,000. In the first scenario, our PKDR is comparable and even better than the estimator based on $h$. For scenarios with misspecification, the PKIPW estimator and the baselines with only $h_0$ respectively perform well in scenario 2 and scenario 3. Notably, the PKDR can constantly perform well in these scenarios, due to its doubly robustness against model mis-specifications. In scenario 4 where both models of $h_0$ and $q_0$ are misspecified, all methods suffer from a large error. Besides, we can see that the PKIPW method has a large variance in scenario 4, where both estimations of the policy function and $q_0$ can be inaccurate due to mis-specifications (Robins et al., 2007; Jiang et al., 2022). It is worth mentioning that compared to others, DFPV exhibits minimal errors in scenario 4. This could be attributed to their approach of individually fitting each variable's kernel function using different neural networks, thereby enhancing flexibility in their models.

**Sensitivity Analysis.** According to Thm. 6.4, $h_{\mathrm{bw}}$ is the trade-off between bias and variance. To show this, we report the cMSE as $c$ in $h_{\mathrm{bw}} := c\hat{\sigma}_A n^{-1/5}$ varies in $\{0.5, 1.0, 1.5, ..., 4.0\}$. As $c$ (*i.e.*,

$h_{\text{bw}}$) increases, the cMSE first decreases, then rises, and reaches its optimum at $c = 1.5$, which is consistent with the optimal value derived in Kallus et al. (2021).

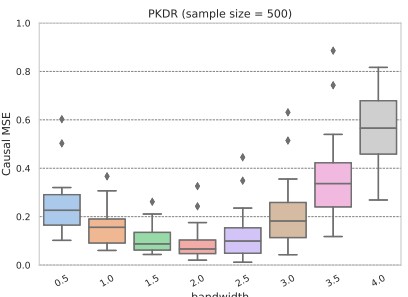 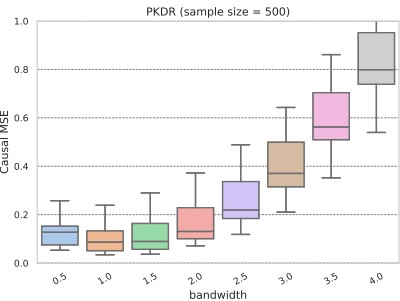

Figure 3: Sensitive analysis of $c$ in $h_{bw} = c\hat{\sigma}_A n^{-1/5}$ in PKIPW (left) and PKDR (right) estimators.

### 7.1.2 HIGH DIMENSIONAL STUDY

**Data generation.** We follow Colangelo & Lee (2020); Singh (2020) to generate data, in which we set $\dim(X) = 100$, $\dim(Z) = 10$, and $\dim(W) = 10$. Specifically, we set $X \sim N(0, \Sigma)$ with $\Sigma \in \mathbb{R}^{100 \times 100}$ has $\Sigma_{ii} = 1$ for $i \in [\dim(X)]$ and $\Sigma_{ij} = \frac{1}{2} \cdot \mathbb{I}|i - j| = 1$ for $i \neq j$. The outcome $Y$ is generated from $Y = A^2 + 1.2A + 1.2(X^\top \beta_x + W^\top \beta_w) + AX_1 + 0.25U$, where $\beta_x, \beta_w$ exhibit quadratic decay, *i.e.*, $[\beta_x]_j = j^{-2}$. More details can be found in the Appx. H.

**Results.** We report the mean and std of cMSE over 20 times with sample sizes set to 1,000 and 2,000, as depicted in Fig. 2 and Tab. 1. As shown, we find that the ATE curve fitted by PKDR estimator is closest to the real curve, and its cMSE is also the lowest. This result suggests the robustness of our methods against high-dimensional covariates.

### 7.2 LEGALIZED ABORTION AND CRIME

We obtain the data from Donohue III & Levitt (2001); Mastouri et al. (2021) that explores the relationship between legalized abortion and crime. In this study, we take the treatment as the effective abortion rate, the outcome variable $Y$ as the murder rate, the treatment-inducing proxy $Z$ as the generosity towards families with dependent children, and the outcome-inducing proxies $W$ as beer consumption per capita, log-prisoner population per capita, and concealed weapons laws. We follow the protocol Woody et al. (2020) to preprocess data. We take the remaining variables as the unobserved confounding variables $U$. Following Mastouri et al. (2021), the ground-truth value of $\beta(a)$ is taken from the generative model fitted to the data.

**Results.** The results are presented in Fig. 2 and Tab. 1. It is evident that all three methods effectively estimate $\beta(a)$, which suggests the utility of our method in real-world scenarios. However, when $a$ falls within the range of $[1.5, 2]$, deviations become apparent in the fitted curve. We attribute these deviations to an inadequate sample size as Fig. 2. It's worth noting that the DFPV method employing Neural Networks (NN) exhibits higher variances. This suggests potential numerical instability in certain experiments, a phenomenon in line with observations made in Kompa et al. (2022).

## 8 CONCLUSION

In this paper, we propose a kernel-based doubly robust estimator for continuous treatments within the proximal causal framework, where we replace the conventional indicator function with a kernel function. Additionally, we propose a more efficient approach to estimating the nuisance function $q_0$ by estimating the policy function and incorporating it into a min-max optimization. Our analysis reveals that the MSE converges at a rate of $O(n^{-4/5})$ when we select the optimal bandwidth to balance bias and variance. We demonstrate the utility of our PKDR estimator in synthetic as well as the legalized abortion and crime dataset.

**Limitation and future works.** Our estimator is required to estimate the policy function, which may lead to a large variance especially when the policy function is mis-specified. Potential solutions include the variance reduction method including the stabilized IPW estimator, whose estimation forms and theoretical analysis will be explored in the future.

## 9 ACKNOWLEDGMENTS

This work was supported by the National Key Research and Development Program of China (No. 2022YFC2405100); STI 2030—Major Projects (No. 2022ZD0205300); Shanghai Municipal Science and Technology Major Project (No.2021SHZDZX0103); the State Key Program of National Natural Science Foundation of China under Grant No. 12331009. The computations in this research were performed using the CFFF platform of Fudan University.

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

APPENDIX

# A  PRELIMINARIES

## A.1  NOTATION

In this section, we will define some notations used throughout the proof in the appendix. Moreover, we will introduce other notations in the corresponding subsection.

Table 2: Table of Notations

| Notation | Meaning |
|---|---|
| $Z, W$ | Treatment-inducing proxy and outcome-inducing proxy |
| $X, U$ | Covariates and unobserved confounders |
| $Y(a)$ | The potential outcome with $A = a$ |
| $\mathrm{do}(a)$ | $\mathrm{do}(A = a)$, intervention variable $A_S$ with a value of $a_S$ |
| $\beta(a)$ | ATE at point $a$ obtained by Eq. 1 or 4 
 Average causal effect at point $a$ |
| $\hat{\beta}(a)$ | Estimate of $\beta(a)$ |
| $h_0 := h_0(a, w, x)$ | The nuisance/bridge function that satisfies Eq. 2 |
| $q_0 := h_0(a, z, x)$ | The nuisance/bridge function that satisfies Eq. 3 |
| $K(\cdot)$ | Kernel function |
| $\kappa_2(K)$ | $\int u^2 k(u) du$ |
| $\Omega_2^{(i)}(K)$ | Second moment of $K$, i.e. $\int (k^{(i)}(u))^2 du$ |
| $K_{h_{\mathrm{bw}}}(a)$ | Kernel function with bw, i.e. $K_{h_{\mathrm{bw}}}(u) = h_{\mathrm{bw}}^{-1} k(u/h_{\mathrm{bw}})$ |
| $\mathbb{P}_O^{h_{\mathrm{bw}}}(\cdot)$ | The distribution approach a point mass at $O$ at $h_{\mathrm{bw}} \to 0$ |
| $\mathbb{P}_O^{\varepsilon h_{\mathrm{bw}}}(\cdot)$ | The special submodel, i.e. $\mathbb{P}_O^{\varepsilon h_{\mathrm{bw}}} = (1 - \varepsilon)\mathbb{P}_O^0 + \varepsilon \mathbb{P}_O^{h_{\mathrm{bw}}}$ |
| $\mathbb{P}_O^0(\cdot)$ | The true distribution |
| $\beta(\cdot; \mathbb{P}^{\varepsilon h_{\mathrm{bw}}})$ | A statistical functional that map the distribution $\mathbb{P}^{\varepsilon h_{\mathrm{bw}}}$ to a real |
| $\mathcal{Q}, \mathcal{H}$ (*resp.* $\mathcal{M}, \mathcal{G}$) | The bridge functional class (*resp.* The dual functional class) |
| $\mathrm{proj}_q(\cdot)$ | Projection operator, i.e. $\mathbb{E}[\cdot \mid A, W, X]$ |
| $\widehat{\mathcal{R}}_n(\delta; \mathcal{G})$ | The empirical Rademacher complexity of the function class $\mathcal{G}$ |
| $\overline{\mathcal{R}}_n(\delta; \mathcal{G})$ | The population Rademacher complexity of the function class $\mathcal{G}$ |
| $\delta_n$ | The upper bound on the Rademacher complexity of the function class $\mathcal{G}$ |
| $N(\varepsilon, \mathcal{G}, \|\cdot\|_\infty)$ | The size of the smallest empirical cover of $\mathcal{G}$ |
| $H(\epsilon, \mathcal{G}, \|\cdot\|)$ | The empirical metric entropy of $\mathcal{G}$ |
| $\Phi_q, \Phi_h$ | Moment restriction, defined by Eq. 14 |
| $\Phi_q^{\lambda_m}, \Phi_h^{\lambda_g}$ | Moment restriction with regularization , defined by Eq. 12 |
| $\Phi_q^n, \Phi_h^n$ | The empirical version of $\Phi_q, \Phi_h$ |
| $\Phi_q^{n, \lambda_m}, \Phi_h^{n, \lambda_g}$ | The empirical version of $\Phi_q^{\lambda_m}, \Phi_h^{\lambda_g}$ |
| $\mathbb{E}[\cdot], \mathbb{P}[\cdot]$ | The expectation and the probability distribution of a random variable |
| $\|f\|_{L_2}$ | $\sqrt{\int |f(x)|^2 \, d\mathbb{P}(x)}$ |
| $\|x\|_2$ | $\sqrt{\sum_i |x_i|^2}$ |

## A.2 CRITICAL RADIUS

The bound we provide is based on the critical radii of the function classes involved, as described in Sec. 6. These critical radius are defined in terms of the empirical localized Rademacher critical radius, which characterizes the critical radius of a function class up to a constant factor. Specifically, the empirical Rademacher complexity and population Rademacher complexity of a function class $\mathcal{G} : \mathcal{V} \to [-1, 1]$ is defined as follows:

$$\widehat{\mathcal{R}}_n(\delta; \mathcal{G}) = \mathbb{E}_{\{\epsilon_i\}_{i=1}^n} \left[ \sup_{g \in \mathcal{G}, \|g\|_2 \leq \delta} \left| \frac{1}{n} \sum_{i=1}^n \epsilon_i g(v_i) \right| \right]$$

$$\overline{\mathcal{R}}_n(\delta; \mathcal{G}) = \mathbb{E}_{\{\epsilon_i\}_{i=1}^n, \{v_i\}_{i=1}^n} \left[ \sup_{g \in \mathcal{G}, \|g\|_2 \leq \delta} \left| \frac{1}{n} \sum_{i=1}^n \epsilon_i g(v_i) \right| \right]$$

where $\{v_i\}_{i=1}^n$ are i.i.d. samples from some distribution $D$ on $\mathcal{V}$ and $\{\epsilon_i\}_{i=1}^n$ are i.i.d. Rademacher random variables taking values equiprobably in $\{-1, 1\}$. The empirical critical radius is defined as any solution $\hat{\delta}_n$ to $\widehat{\mathcal{R}}_n(\delta; \mathcal{G}) \leq \delta^2$ and the critical radius of a function class $\mathcal{G}$ is the smallest solution $\delta_n^*$ to the inequality $\overline{\mathcal{R}}_n(\delta; \mathcal{G}) \leq \delta^2$. Proposition 14.1 of Wainwright (2019) shows that w.p. $1 - \zeta$,

$$\delta_n = O\left( \hat{\delta}_n + \sqrt{\frac{\log(1/\zeta)}{n}} \right).$$

Thus we can choose $\delta_n$ based on the empirical critical radius $\hat{\delta}_n$. Moreover, Dikkala et al. (2020) suggests using the covering number to obtain an upper bound on the empirical Rademacher complexity and thus the critical radius. We denote with $N(\epsilon, \mathcal{G}, \|\cdot\|)$ as the size of the smallest empirical-cover of $\mathcal{G}$. The empirical metric entropy of $\mathcal{G}$ is defined as $H(\epsilon, \mathcal{G}, \|\cdot\|) = \log(N(\epsilon, \mathcal{G}, \|\cdot\|))$. An empirical $\delta$-slice of $\mathcal{G}$ is defined as $\mathcal{G}_\delta = \{g \in \mathcal{G} : \|g\|_{2,n} \leq \delta\}$. Then the empirical critical radius of $\mathcal{G}$ is upper bounded by any solution to the inequality:

$$\int_{\delta^2/8}^{\delta} \sqrt{\frac{H(\epsilon, \mathcal{G}_\delta, \|\cdot\|)}{n}} d\epsilon \leq \frac{\delta^2}{20} \tag{10}$$

## B    RELATED WORKS

Regarding the proximal causal inference framework, there is a growing literature on using proxy variables for causal inference from observational data, due to its ability to account for unmeasured $U$ through two confounding proxy variables: a treatment-inducing proxy $Z$ and an outcome-inducing proxy $W$, which are respectively independent to the outcome and the treatment.

The proximal causal learning (PCL) can be dated back to Kuroki & Pearl (2014), which established the identification of causal effects in the presence of unobserved confounders under linear models. Then Miao et al. (2018b) and its extensions (Shi et al., 2020; Tchetgen et al., 2020) proposed to leverage two proxy variables $(Z, W)$ for causal identification by estimating the outcome bridge function (Eq. 2). Building upon this foundation, Cui et al. (2023); Ying et al. (2023) introduced a treatment bridge function (Eq. 3) and incorporated it into the Proximal Inverse Probability Weighting (PIPW) estimator. Besides, under binary treatments, they derived the Proximal Doubly Robust (PDR) estimator via influence functions. However, their methods cannot handle continuous treatments, whereas the treatments can be continuous in many real-world scenarios, including social science, biology, and economics. The main challenge for continuous treatments lies in the estimation infeasibility caused by the indicator function in two estimators. **To the best of our knowledge, we are the first to generalize the PIPW estimator and PDR estimator to the continuous case by replacing the conventional indicator function with a kernel function.** The advantage of using the kernel function is that we do not need to discretize the continuous data, but use the kernel function to incorporate local information about the similar treatments. Besides,**we derive the corresponding influence function, a process that involves handling the Gateaux derivative of bridge functions.**

Most existing work focuses on how to estimate the outcome bridge function. Singh (2020) and Mastouri et al. (2021) propose to use a two-stage kernel estimator for the outcome bridge function (**KPV**), and Xu et al. (2021) further improved upon this with an adaptive features derived from neural networks (**DFPV**). Besides, an alternative approach based on maximum moment restriction (MMR) uses single-stage estimators of the bridge function. The masterpiece in this regard is that Mastouri et al. (2021) extends the MMR framework to the proximal setting through the use of kernel functions (**PMMR**). Kompa et al. (2022) introducex data-adaptive kernel functions derived from neural networks. Another traditional method is to transform the conditional moment equation into an unconditional moment equation, and then solve a minimax optimization problem (Dikkala et al., 2020; Ghassami et al., 2022; Qi et al., 2023) (**MINIMAX, POR**). However, these methods that estimate only the outcome bridge function, rather than also estimating the treatment bridge function, which would permit us to construct a doubly robust estimator. For the treatment bridge function, (Kallus et al., 2021; Ghassami et al., 2022) solved $q_0(a, Z, X)$ by running an optimization algorithm for each $a = 0, 1$. However, it is computationally infeasible for continuous treatments. Instead of running an optimization for each $a$, we would like to estimate $q_0(A, Z, X)$ with a single optimization algorithm. **To achieve this goal, we propose a two-stage estimation algorithm. We first estimate the policy function and then convert the conditional moments into equivalent forms, which can lead to a min-max optimization problem.**

## C  REGULARITY CONDITION

One conventional approach to studying their solutions is through singular value decomposition, as discussed by Carrasco et al. (2007). We first introduce the singular value decomposition of the operator:

Given Hilbert spaces $H_1$ and $H_2$, a compact operator $K : H_1 \longmapsto H_2$ and its adjoint operator $K' : H_2 \longmapsto H_1$, there exists a singular system $(\lambda_n, \varphi_n, \psi_n)_{n=1}^{+\infty}$ of $K$ with nonzero singular values $\{\lambda_n\}$ and orthogonal sequences $\{\varphi_n \in H_1\}$ and $\{\psi_n \in H_2\}$ such that

$$K\varphi_n = \lambda_n \psi_n, K'\psi_n = \lambda_n \varphi_n.$$

By means of singular value decomposition, Picard's theorem characterizes the conditions for the existence of solutions of the corresponding Fredholm integral equations of the first type. We apply Picard's theorem to the setting of proxy variables.

Let $L^2\{\mathbb{P}(x)\}$ denote the space of all square integrable functions of $x$ with respect to a cumulative distribution function $\mathbb{P}(x)$, which is a Hilbert space with inner product $\langle g_1, g_2 \rangle = \int g_1(x) g_2(x) \mathrm{d}\mathbb{P}(x)$. For brevity, we replace $W_{ij}$ and $Z_{ij}$ with $W$ and $Z$ below. Let $T_{a,x}$ denote the operator: $L^2\{\mathbb{P}(w|a,x)\} \to L^2\{\mathbb{P}(z|a,x)\}$, $T_{a,x}h = \mathbb{E}[h(W)|z,a,x]$ and let $(\lambda_{a,x,n}, \varphi_{a,x,n}, \phi_{a,x,n})_{n=1}^{\infty}$ denote a singular value decomposition of $T_{a,x}$. Also let $T'_{a,x}$ denote the operator: $L^2\{\mathbb{P}(z|a,x)\} \to L^2\{\mathbb{P}(w|a,x)\}$, $T'_{a,x}q = \mathbb{E}[q(Z)|w,a,x]$ and let $(\lambda'_{a,x,n}, \varphi'_{a,x,n}, \phi'_{a,x,n})_{n=1}^{\infty}$ denote a singular value decomposition of $T'_{a,x}$. We assume the following regularity conditions:

**Assumption C.1.** *(Regularity conditions)*

1. *(Existence of compact operator $T_{a,x}$ and $T'_{a,x}$)*

$$\int \int p(w|z,a,x) p(z|w,a,x) \mathrm{d}w \mathrm{d}z < \infty;$$

2. *(Existence of solutions)*

$$\int \mathbb{E}^2[Y|z,a,x] p(z|a,x) \mathrm{d}z < \infty;$$

3. *(Eigenvalue structure of compact operator $T_{a,x}$)*

$$\sum_{n=1}^{\infty} \lambda_{a,x,n}^{-2} |\langle \mathbb{E}[Y|z,x,x], \phi_{a,x,n} \rangle|^2 < \infty;$$

4. *(Existence of solutions)*

$$\int p^{-2}(a|w,x) p(w|a,x) \mathrm{d}w < \infty;$$

5. *(Eigenvalue structure of compact operator $T'_{a,x}$)*

$$\sum_{n=1}^{\infty} \lambda'^{-2}_{a,x,n} |\langle p^{-1}(a|w,x), \phi'_{a,x,n} \rangle|^2 < \infty.$$

**Theorem C.2** (Cui et al. (2023)). *Under Assumption C.1(1,2,3) and Assumption 3.4(1), there exist functions $h(w, a, x)$ such that*

$$\mathbb{E}[Y|Z, A, x] = \int h(w, A, x) \mathrm{d}\mathbb{P}(w|Z, A, x),$$

*almost surely.*

**Theorem C.3** (Cui et al. (2023)). *Under Assumption C.1(1,4,5) and Assumption 3.4(2), there exist functions $q(z, a, x)$ such that*

$$\mathbb{E}[q(Z, a, x)|W, A = a, x] = \frac{1}{p(A = a|W, x)}$$

*almost surely.*

## D  ESTIMATING NUISANCE FUNCTION

We next return to how to solve the nuisance function $q$. For simplicity, define

$$\mathcal{R}_q(q; p) = \mathbb{E}\left[\left(\frac{1}{p(A \mid W, X)} - q(A, Z, X)\right) \mid A, W, X\right]$$

Notice that $\mathcal{R}_q(q, p)$ being zero almost surely is actually a conditional moment equation, which is equivalent to finding $q$ such that the following residual mean squared error (RMSE) is minimized for all $q$:

$$\min_{q \in \mathcal{Q}} \mathcal{L}_q(q; p) := \mathbb{E}\left[\left(\mathcal{R}_q(q, p)\right)^2\right] \tag{11}$$

### D.1  PROOF OF LEMMA 5.1

In the following steps, we will utilize the technique of minimax estimation. To begin with, we will introduce Interchangeability:

**Definition D.1** (Fenchel duality). *Let $\ell : \mathbb{R} \times \mathbb{R} \to \mathbb{R}_+$ be a proper, convex, and lower semi-continuous loss function for any value in its first argument and $\ell_y^\star := \ell^\star(y, \cdot)$ a convex conjugate of $\ell_y := \ell(y, \cdot)$ which is also proper, convex, and lower semi-continuous w.r.t. the second argument. Then, $\ell_y(v) = \max_u\{uv - \ell_y^\star(u)\}$. The maximum is achieved at $v \in \partial\ell^\star(u)$, or equivalently $u \in \partial\ell(v)$.*

**Theorem D.2** (Interchangeability). *Let $\omega$ be a random variable on $\Omega$ and, for any $\omega \in \Omega$, the function $f(\cdot, \omega) : \mathbb{R} \to (-\infty, \infty)$ is proper and upper semi-continuous concave function. Then,*

$$\mathbb{E}_\omega\left[\max_{u \in \mathbb{R}} f(u, \omega)\right] = \max_{u(\cdot) \in \mathcal{U}(\Omega)} \mathbb{E}_\omega[f(u(\omega), \omega)],$$

*where $\mathcal{U}(\Omega) := \{u(\cdot) : \Omega \to \mathbb{R}\}$ is the entire space of functions defined on the support $\Omega$.*

**Lemma 5.1.** *Denote $\|f(X)\|_{L_2}^2 := \mathbb{E}[f^2(X)]$. For any parameter $\lambda_m > 0$, we have*

$$\mathcal{L}_q(q; p) = \sup_{m \in \mathcal{M}} \mathbb{E}\left[m(A, W, X)\left(q_0(A, Z, X) - 1/p(A|W, X)\right)\right] - \lambda_m\|m(A, W, X)\|_{L_2}^2,$$

*where $\mathcal{M}$ is the space of continuous functions over $(A, W, X)$.*

*Proof.* Notice that the squared loss function $\ell_y(v) = \frac{1}{4\lambda}(y - v)^2$, we have $\ell_y^\star(u) = uy + \lambda u^2$. Then by Fenchel duality, we have

$$\begin{aligned}
\ell_y(v) = \frac{1}{4\lambda}(y - v)^2 &= \max_{u \in \mathbb{R}}\left\{vu - \ell_y^\star(u)\right\} \\
&= \max_{u \in \mathbb{R}}\left\{vu - uy - \lambda u^2\right\} = \max_{u \in \mathbb{R}}\left\{(v - y)u - \lambda u^2\right\}
\end{aligned}$$

Let $y = \mathbb{E}\left[\frac{1}{p(A|W,X)} \mid A, W, X\right], v = \mathbb{E}[q(A, Z, X) \mid A, W, X], u = m$, we have

$$\begin{aligned}
&\frac{1}{4\lambda_m}\left(\mathbb{E}\left[\frac{1}{p(A \mid W, X)} \mid A, W, X\right] - \mathbb{E}[q(A, Z, X) \mid A, W, X]\right)^2 \\
&= \max_{m \in \mathbb{R}}\left\{\left(\mathbb{E}[q(A, Z, X) \mid A, W, X] - \mathbb{E}\left[\frac{1}{p(A \mid W, X)} \mid A, W, X\right]\right)m - \lambda_m m^2\right\}
\end{aligned}$$

Therefore, applying the interchangeability, we have

$$
\mathcal{L}_q(q; p) = 4\lambda_m \mathbb{E}\left[\frac{1}{4\lambda_m}\left(\mathbb{E}\left[\left(\frac{1}{p\left(A \mid W, X\right)} - q\left(A, Z, X\right)\right) \mid A, W, X\right]\right)^2\right]
$$

$$
= 4\lambda_m \mathbb{E}\left[\max_{m \in \mathbb{R}}\left\{\left(\mathbb{E}\left[q\left(A, Z, X\right) \mid A, W, X\right] - \mathbb{E}\left[\frac{1}{p\left(A \mid W, X\right)} \mid A, W, X\right]\right) m - \lambda_m m^2\right\}\right]
$$

$$
= 4\lambda_m \mathbb{E}\left[\max_{m \in \mathbb{R}}\left\{\mathbb{E}_Z\left[q\left(A, Z, X\right) - \frac{1}{p\left(A \mid W, X\right)} \mid A, W, X\right] m - \lambda_m m^2\right\}\right]
$$

$$
= 4\lambda_m \max_{m \in \mathcal{M}} \mathbb{E}\left[\mathbb{E}\left[q\left(A, Z, X\right) - \frac{1}{p\left(A \mid W, X\right)} \mid A, W, X\right] m\left(A, W, X\right) - \lambda_m m^2\left(A, W, X\right)\right]
$$

$$
= 4\lambda_m \max_{m \in \mathcal{M}} \mathbb{E}\left[\left(q\left(A, Z, X\right) - \frac{1}{p\left(A \mid W, X\right)}\right) m\left(A, W, X\right) - \lambda_m m^2\left(A, W, X\right)\right]
$$

We define

$$
\Phi_q^{\lambda_m}(q, m; p) = \mathbb{E}\left[\left(q\left(A, Z, X\right) - \frac{1}{p\left(A \mid W, X\right)}\right) m\left(A, W, X\right)\right] - \lambda_m \|m\|_2^2. \tag{12}
$$

As long as the dual function class $\mathcal{M}$ is expressive enough such that $\frac{1}{2\lambda_m}\mathcal{R}_q\left(q, p\right) \in \mathcal{M}$, we have

$$
\mathcal{L}_q(q; p) = \max_{m \in \mathcal{M}} \Phi_q^{n,\lambda_m}(q, m; p)
$$

$\square$

we can express Eq. 11 in an alternative form:

$$
\min_{q \in \mathcal{Q}} \max_{m \in \mathcal{M}} \Phi_q^{\lambda_m}(q, m; p). \tag{13}
$$

We denote the empirical version

$$
\Phi_q^{n,\lambda_m}(q, m; p) = \frac{1}{n}\sum_i \left(q\left(z_i, a_i, x_i\right) - \frac{1}{p\left(a_i \mid w_i, x_i\right)}\right) m\left(a_i, w_i, x_i\right) - \lambda_m \|m\|_{2,n}^2.
$$

Furthermore, for simplicity, we define $\Phi_q(q, m; p)$ as the non regularized version of $\Phi_q^{\lambda_m}$.

$$
\Phi_q(q, m; p) = \mathbb{E}\left[\left(q\left(A, Z, X\right) - \frac{1}{p\left(A \mid W, X\right)}\right) m\left(A, W, X\right)\right] \tag{14}
$$

In fact, due to the fact that we need to estimate the density function $p$, we consider the following min-max optimization problem:

$$
\min_{q \in \mathcal{Q}} \max_{m \in \mathcal{M}} \Phi_q^{n,\lambda_m}(q, m; \hat{p}) \tag{15}
$$

Similarly, for the nuisance function $h$, we consider the following min-max optimization problem:

$$
\min_{h \in \mathcal{H}} \max_{g \in \mathcal{G}} \Phi_h^{n,\lambda_g}(h, g) = \frac{1}{n}\sum_i \left(y_i - h\left(w_i, a_i, x_i\right)\right) g\left(a_i, z_i, x_i\right) - \lambda_g \|g\|_{2,n}^2. \tag{16}
$$

### D.2 COMPARISON OF EXISTING METHODS

Closely related to us is the estimation of the general treatments proposed by Kallus et al. (2021). They hope to estimate the generalized average causal effect (GACE):

$$
J = \mathbb{E}\left[\int Y(a)\pi(a \mid X)\mathrm{d}\mu(a)\right].
$$

where $\pi(a \mid X)$ is contrast function. Based on the idea, they propose the method for solve the nuisance function $q$

$$
\min_{q \in \mathcal{Q}} \max_{m \in \mathcal{M}} \mathbb{E}_n[\pi(A \mid X)q(A, Z, X)m(A, W, X) - (\mathcal{T}m)(W, X)] - \lambda_m \|m\|_{2,n}^2
$$

where $(\mathcal{T}m)(w,x) = \int m(a,w,x)\pi(a|x)d\mu(a)$.

However, consider the continuous treatments $a \in \text{supp}(A)$, then in this case, $\pi(a \mid X) = \mathbb{I}(A = a)$. Correspondingly, the conditional moment equation for the action bridge function $q$ is equivalent to

$$\min_{q \in \mathcal{Q}} \max_{m \in \mathcal{M}} \mathbb{E}_n[\mathbb{I}(A = a)\, q(A, Z, X)m(A, W, X) - (\mathcal{T}m)(W, X)] - \lambda_m \|m\|_{2,n}^2$$

As mentioned above, the main challenge for continuous treatments lies in the estimation infeasibility caused by the indicator function. Therefore we cannot estimate the nuisance function $q$ when the treatment is continuous.

The second paper to solve $q$ is from Ghassami et al. (2022). They estimate the causal effects of binary treatments and illustrate how the double robustness property of these influence functions can be used to formulate estimating equations for the nuisance functions. Specifically,

$$\min_{q \in \mathcal{Q}} \max_{m \in \mathcal{M}} \mathbb{E}_n \left[ \{-\mathbb{I}(A = a)q(Z, X) + 1\} m(W, X) - m^2(W, X) \right]$$

Then functions $\hat{q}(a, z, x) = \mathbb{I}(a = 0)\hat{q}_0(z, x) + \mathbb{I}(a = 1)\hat{q}_1(z, x)$. Note that since the indicator function appears in their optimization equation, we still cannot solve the case of continuous treatments.

Instead of running an optimization for each $a$, we would like to estimate $q_0(A, Z, X)$ with a single optimization algorithm. To achieve this goal, we propose a two-stage estimation algorithm. We first estimate the policy function and then convert the conditional moments into equivalent forms, which can lead to a min-max optimization problem.

$$\min_{q \in \mathcal{Q}} \max_{m \in \mathcal{M}} \frac{1}{n} \sum_i \left( q(a_i, z_i, x_i) - \frac{1}{p(a_i|w_i, x_i)} \right) m(a_i, w_i, x_i) - \lambda_m \|m\|_{2,n}^2.$$

Since the treatment is continuous, we have to estimate the policy function before solving the moment equation. However, when the treatment is binary, we can transform the moment equation so that the policy function disappears. This is the cost of estimating the causal effects of continuous treatments.

# E PROOFS AND DERIVATION

## E.1 PROOF OF THEOREM 4.2

**Theorem 4.2.** *Under assump. 4.1, suppose $\beta(a) = \mathbb{E}[\mathbb{I}(A = a)q_0(a, Z, X)Y]$ is continuous and bounded uniformly respect to a, then we have*

$$\mathbb{E}[Y(a)] = \mathbb{E}[\mathbb{I}(A = a)q_0(a, Z, X)Y] = \lim_{h_{\mathrm{bw}} \to 0} \mathbb{E}\left[K_{h_{\mathrm{bw}}}(A - a)q_0(a, Z, X)Y\right],$$

*Proof.* By definition we have

$$
\begin{aligned}
\mathbb{E}[\mathbb{I}(A = a_0)q_0(a_0, Z, X)Y] &= \int_{O\backslash A} yq(a_0, z, x)\mathrm{d}\mathbb{P}\left(a_0, z, x, y\right) \\
&= \int_O \delta_{a_0}\left(a\right) yq_0(a, z, x)\mathrm{d}\mathbb{P}\left(a, z, x, y\right) \\
&= \int_A \delta_{a_0}\left(a\right) \int_{O\backslash A} yq_0(a, z, x)p\left(a, z, x, y\right)\mathrm{d}\mu(z, x, y)\mathrm{d}a \\
&= \int_A \delta_{a_0}\left(a\right) \beta\left(a\right)\mathrm{d}a = \langle\delta_{a_0}, \beta\rangle = \beta\left(a_0\right)
\end{aligned}
$$

where the last equation uses the properties of the Dirac function. Similarly for the kernel function, we also have

$$\langle K_{h_{\mathrm{bw}}}, \beta\rangle = \int_A K_{h_{\mathrm{bw}}}(a - a_0)\beta\left(a\right)\mathrm{d}a \xrightarrow{a = h_{\mathrm{bw}}s + a_0} \int_S K(s)\beta\left(h_{\mathrm{bw}}s + a_0\right)\mathrm{d}s$$

As $\beta$ is continous and bounded uniformly, this integral is dominated by $CK(s)$. Moreover, because $\beta$ is continuous, the integral converges point-wise to $K(s)\beta(a_0)$. Applying the dominated convergence theorem and Assumption 4.1(The kernel function integral is 1.) yields

$$\lim_{h_{\mathrm{bw}} \to 0} \langle K_{h_{\mathrm{bw}}}, \beta\rangle = \int_S \lim_{h_{\mathrm{bw}} \to 0} K(s)\beta\left(h_{\mathrm{bw}}s + a_0\right)\mathrm{d}s = \int_S K(s)\beta\left(a_0\right)\mathrm{d}s = \beta\left(a_0\right)$$

which finally shows

$$\mathbb{E}[Y(a)] = \lim_{h_{\mathrm{bw}} \to 0} \mathbb{E}\left[K_{h_{\mathrm{bw}}}(A - a)q_0(A, Z, X)Y\right]$$

$\square$

## E.2 PROOF OF THEOREM 4.5

We denote $\mathbb{P}_X$ as the distribution function for any variable $X$, and rewrite $\beta(a)$ as $\beta(a; \mathbb{P}_O^0)$ where $\mathbb{P}_O^0$ denotes the true distribution for $O := (A, Z, W, X, Y)$. Besides, we consider the special sub-model $\mathbb{P}_O^{\varepsilon h_{\mathrm{bw}}} = (1-\varepsilon)\mathbb{P}_O^0 + \varepsilon\mathbb{P}_O^{h_{\mathrm{bw}}}$, where $\mathbb{P}_O^{h_{\mathrm{bw}}}(\cdot)$ maps a point $o$ to a distribution of $O$, *i.e.*, $\mathbb{P}_O^{h_{\mathrm{bw}}}(o)$ for any $o$ denotes the distribution of $O$ that approach a point mass at $o$ as $h_{\mathrm{bw}} \to 0$.

**Theorem 4.5.** *Under a nonparametric model, the limit of the Gateaux derivative is*

$$\lim_{h_{\mathrm{bw}} \to 0} \frac{\partial}{\partial\varepsilon}\beta(a; \mathbb{P}^{\varepsilon h_{\mathrm{bw}}})\bigg|_{\varepsilon=0} = (Y - h_0(a, W, X))\, q_0(a, Z, X)\lim_{h_{\mathrm{bw}} \to 0} K_{h_{\mathrm{bw}}}(A - a) + h_0\left(a, W, X\right) - \beta(a)$$

*Proof.* Similar to $\beta(a)$ rewritten as $\beta(a; \mathbb{P}_O^0)$, we can rewrite $h_0\left(a, w, x\right)$ as $h_0\left(a, w, x; \mathbb{P}_{AWX}^0\right)$. For simplicity, we omit the subscript of the distribution. Please identify according to context.

$$
\begin{aligned}
\frac{\partial}{\partial\varepsilon}\beta(a; \mathbb{P}^{\varepsilon h_{\mathrm{bw}}})\bigg|_{\varepsilon=0} &= \frac{\partial}{\partial\varepsilon}\int h_0\left(a, w, x; \mathbb{P}^{\varepsilon h_{\mathrm{bw}}}\right)\mathrm{d}\mathbb{P}^{\varepsilon h_{\mathrm{bw}}}\left(w, x\right) \\
&= \underbrace{\int \frac{\partial}{\partial\varepsilon}\, h_0\left(a, w, x; \mathbb{P}^{\varepsilon h_{\mathrm{bw}}}\right)\big|_{\varepsilon=0}\mathrm{d}\mathbb{P}^0\left(w, x\right)}_{\textbf{(I)}} \\
&\quad + \underbrace{\int h_0\left(a, w, x\right)\frac{\partial}{\partial\varepsilon}\, p^{\varepsilon h_{\mathrm{bw}}}\left(w, x\right)\big|_{\varepsilon=0}\mathrm{d}\mu\left(w, x\right)}_{\textbf{(II)}}
\end{aligned}
$$

Since $\mathbb{P}_O^{\varepsilon h_{\mathrm{bw}}} = (1 - \varepsilon)\mathbb{P}_O^0 + \varepsilon\mathbb{P}_O^{h_{\mathrm{bw}}}$, we have

$$\frac{\partial}{\partial\varepsilon}\left.\mathbb{P}_O^{\varepsilon h_{\mathrm{bw}}}\right|_{\varepsilon=0} = \mathbb{P}_O^{h_{\mathrm{bw}}} - \mathbb{P}_O^0.$$

For term **(II)**

$$\textbf{(II)} = \int h_0\left(a, w, x\right)\frac{\partial}{\partial\varepsilon}\left.p^{\varepsilon h_{\mathrm{bw}}}\left(w, x\right)\right|_{\varepsilon=0}\mathrm{d}\mu\left(w, x\right)$$

$$= \int h_0\left(a, w, x\right)\left(p^{h_{\mathrm{bw}}}\left(w, x\right) - p^0\left(w, x\right)\right)\mathrm{d}\mu\left(w, x\right)$$

$$\xrightarrow{h_{\mathrm{bw}}\to 0} h_0\left(a, W, X\right) - \beta\left(a\right)$$

For term **(I)**

$$\textbf{(I)} = \int\frac{\partial}{\partial\varepsilon}\left.h_0\left(a, w, x; \mathbb{P}^{\varepsilon h_{\mathrm{bw}}}\right)\right|_{\varepsilon=0}\mathrm{d}\mathbb{P}^0\left(w, x\right)$$

$$= \int\frac{\partial}{\partial\varepsilon}\left.h_0\left(a, w, x; \mathbb{P}^{\varepsilon h_{\mathrm{bw}}}\right)\right|_{\varepsilon=0}\frac{p^0\left(a, w, x\right)}{p^0\left(a \mid w, x\right)}\mathrm{d}\mu\left(w, x\right)$$

$$= \int\frac{\partial}{\partial\varepsilon}\left.h_0\left(a, w, x; \mathbb{P}^{\varepsilon h_{\mathrm{bw}}}\right)\right|_{\varepsilon=0}p^0\left(a, w, x\right)\int q_0\left(a, z, x\right)p^0\left(z \mid a, w, x\right)\mathrm{d}\mu z\mathrm{d}\mu\left(w, x\right)$$

$$= \int q_0\left(a, z, x\right)\frac{\partial}{\partial\varepsilon}\left.h_0\left(a, w, x; \mathbb{P}^{\varepsilon h_{\mathrm{bw}}}\right)\right|_{\varepsilon=0}p^0\left(w, y \mid a, z, x\right)p^0\left(a, z, x\right)\mathrm{d}\mu\left(w, x, z, y\right)$$

And by Eq. 2, we have

$$\left.\frac{\partial}{\partial\varepsilon}\int\left(y - h_0\left(a, w, x; \mathbb{P}^{\varepsilon h_{\mathrm{bw}}}\right)\right)p^{\varepsilon h_{\mathrm{bw}}}\left(y, w \mid a, z, x\right)\mathrm{d}\mu\left(y, w\right)\right|_{\varepsilon=0} = 0$$

Then

$$\int\frac{\partial}{\partial\varepsilon}\left.h_0\left(a, w, x; \mathbb{P}^{\varepsilon h_{\mathrm{bw}}}\right)\right|_{\varepsilon=0}p^0\left(w, y \mid a, z, x\right)\mathrm{d}\mu\left(w, y\right)$$

$$= \int\left[y - h_0\left(a, w, x\right)\right]\frac{\frac{\partial}{\partial\varepsilon}p^{\varepsilon h_{\mathrm{bw}}}\left.\left(w, y, a, z, x\right)\right|_{\varepsilon=0}}{p^0\left(a, z, x\right)}\mathrm{d}\mu\left(w, y\right)$$

$$- \int\left[y - h_0\left(a, w, x\right)\right]\frac{p^0\left(w, y, a, z, x\right)\frac{\partial}{\partial\varepsilon}p^{\varepsilon h_{\mathrm{bw}}}\left.\left(a, z, x\right)\right|_{\varepsilon=0}}{(p^0)^2\left(a, z, x\right)}\mathrm{d}\mu\left(w, y\right)$$

where we use the equation

$$\left.\frac{\partial}{\partial\varepsilon}p^{\varepsilon h_{\mathrm{bw}}}\left(w, y \mid a, z, x\right)\right|_{\varepsilon=0} = \frac{\frac{\partial}{\partial\varepsilon}p^{\varepsilon h_{\mathrm{bw}}}\left.\left(w, y, a, z, x\right)\right|_{\varepsilon=0}}{p^0\left(a, z, x\right)} - \frac{p^0\left(w, y, a, z, x\right)\frac{\partial}{\partial\varepsilon}p^{\varepsilon h_{\mathrm{bw}}}\left.\left(a, z, x\right)\right|_{\varepsilon=0}}{(p^0)^2\left(a, z, x\right)}$$

Substituting (**I**), we obtain

$$(\mathbf{I}) = \int q_0\left(a,z,x\right)\left(y - h_0\left(a,w,x\right)\right)p^0\left(a,z,x\right)\frac{\frac{\partial}{\partial\varepsilon}p^{\varepsilon h_{\mathrm{bw}}}\left(w,y,a,z,x\right)\big|_{\varepsilon=0}}{p^0\left(a,z,x\right)}\mathrm{d}\mu\left(w,x,z,y\right)$$

$$- \int q_0\left(a,z,x\right)\left(y - h_0\left(a,w,x\right)\right)p^0\left(a,z,x\right)\frac{p^0\left(o\right)\frac{\partial}{\partial\varepsilon}p^{\varepsilon h_{\mathrm{bw}}}\left(a,z,x\right)\big|_{\varepsilon=0}}{\left(p^0\right)^2\left(a,z,x\right)}\mathrm{d}\mu\left(w,x,z,y\right)$$

$$= \int q_0\left(a,z,x\right)\left(y - h_0\left(a,w,x\right)\right)p^0\left(a,z,x\right)\frac{p^{h_{\mathrm{bw}}}\left(o\right) - p^0\left(o\right)}{p^0\left(a,z,x\right)}\mathrm{d}\mu\left(w,x,z,y\right)$$

$$- \int q_0\left(a,z,x\right)\left(y - h_0\left(a,w,x\right)\right)p^0\left(a,z,x\right)\frac{p\left(o\right)\left(p^{h_{\mathrm{bw}}}\left(a,z,x\right) - p^0\left(a,z,x\right)\right)}{\left(p^0\right)^2\left(a,z,x\right)}\mathrm{d}\mu\left(w,x,z,y\right)$$

$$\xrightarrow{h_{\mathrm{bw}}\to 0} q_0\left(a,Z,X\right)\left(Y - h_0\left(a,w,x\right)\right)\lim_{h_{\mathrm{bw}}\to 0}p_A^{h_{\mathrm{bw}}}\left(a\right)$$

$$+ \int q_0\left(a,z,x\right)\left(y - h_0\left(a,w,x\right)\right)p^0\left(w,y\mid a,z,x\right)p^{h_{\mathrm{bw}}}\left(a,z,x\right)\mathrm{d}\mu\left(w,x,z,y\right)$$

$$\xrightarrow{h_{\mathrm{bw}}\to 0} q_0\left(a,Z,X\right)\left(Y - h_0\left(a,W,X\right)\right)\lim_{h_{\mathrm{bw}}\to 0}p_A^{h_{\mathrm{bw}}}\left(a\right)$$

where the last line is because of Eq. 2.

The corresponding probability density function (pdf) $p_O^{h_{\mathrm{bw}}}(o) = K_{h_{\mathrm{bw}}}(O - o)\mathbb{I}(p_O^0(o) > h_{\mathrm{bw}})$ is our kernel density, and we thus have $\lim_{h_{\mathrm{bw}}\to 0}p_O^{h_{\mathrm{bw}}}(o) = \lim_{h_{\mathrm{bw}}\to 0}K_{h_{\mathrm{bw}}}(O - o)$. Combining the two terms, we get

$$\lim_{h_{\mathrm{bw}}\to 0}\frac{\partial}{\partial\varepsilon}\beta(a;\mathbb{P}^{\varepsilon h_{\mathrm{bw}}})\bigg|_{\varepsilon=0} = (Y - h_0(a,W,X))\,q_0(a,Z,X)\lim_{h_{\mathrm{bw}}\to 0}K_{h_{\mathrm{bw}}}(A - a) + h_0\left(a,W,X\right) - \beta(a)$$

$$\square$$

### E.3 Proof of Theorem 6.2

To prove Theorem 6.2, we first give some Lemma.

**Lemma E.1** (Theorem 14.1 in Wainwright (2019)). *Given a star-shaped, $b$-uniformly bounded function class $\mathcal{F}$, let $\delta_n$ be any positive solution of the inequality $\overline{\mathcal{R}}_n(\delta;\mathcal{G}) \leq \delta^2/b$. Then for any $t \geq \delta_n$, we have*

$$\left|\|f\|_{2,n}^2 - \|f\|_2^2\right| \leq \frac{1}{2}\|f\|_2^2 + \frac{t^2}{2}, \qquad \forall f \in \mathcal{F},$$

*with probability at least $1 - c_1 e^{-c_2 nt^2/b^2}$. If in addition $n\delta_n^2 \geq 2\log\left(4\log\left(1/\delta_n\right)\right)/c_2$, then we have that*

$$\left|\|f\|_{2,n}^2 - \|f\|_2^2\right| \leq c_0\delta_n, \qquad \forall f \in \mathcal{F},$$

*with probability at least $1 - c_1'\exp(-c_2' n\delta_n^2/b^2)$.*

**Lemma E.2** (Lemma 11 in Foster & Syrgkanis (2019)). *Consider a function class $\mathcal{F}$, with $\sup_{f\in\mathcal{F}}\|f\|_\infty \leq b$, and pick any $f^\star \in \mathcal{F}$. Also, consider a loss function $\ell : \mathbb{R} \times \mathcal{Y} \mapsto \mathbb{R}$ which is $L$-Lipschitz in its first argument with respect to the $l_2$ norm. Let $\delta_n^2 \geq \frac{4d\log(41\log(2c_2 n))}{c_2 n}$ be any solution to the inequalities:*

$$\overline{\mathcal{R}_n}\left(\delta;\mathrm{star}\left(\mathcal{F} - f\right)\right) \leq \frac{\delta^2}{\|\mathcal{F}\|_\infty},$$

*where $\mathrm{star}\left(\mathcal{F} - f\right) = \alpha\left(f - f^*\right)$ for $\forall f \in \mathcal{F}, \alpha \in [0,1]$. Then for any $t \geq \delta_n$ and some universal constants $c_1, c_2 > 0$, with probability $1 - c_1 e^{-c_2 nt^2/b^2}$, it holds that*

$$\left|\left(\mathbb{E}_n\left[\ell\left(f\left(x\right),y\right)\right] - \mathbb{E}_n\left[\ell\left(f^\star\left(x\right),y\right)\right]\right) - \left(\mathbb{E}\left[\ell\left(f\left(x\right),y\right)\right] - \mathbb{E}\left[\ell\left(f^\star\left(x\right),y\right)\right]\right)\right| \leq 18Ldt\left\{\|f - f^\star\|_2 + t\right\},$$

*for any $f \in \mathcal{F}$. If furthermore, the loss function $\ell$ is linear in $f$, i.e., $\ell((f + f')(x),y) = \ell(f(x),y) + \ell(f'(x),y)$ and $\ell(\alpha f(x),y) = \alpha\ell(f(x),y)$, then the lower bound on $\delta_n^2$ is not required. If the outcome $\hat{f}$ of constrained ERM satisfies that with the same probability,*

$$\mathbb{E}\left[\ell\left(\hat{f}(x),y\right)\right] - \mathbb{E}\left[\ell\left(f^\star(x),y\right)\right] \leq 18Ldt\left\{\left\|\hat{f} - f^\star\right\|_2 + t\right\}.$$

**Lemma E.3.** *Let some $a, b, d \geq 0$ be given, and suppose that*

$$aX^2 \leq bX + d,$$

*for some $X \geq 0$. Then, we have*

$$X \leq \frac{b + \sqrt{ad}}{a}$$

*Proof.* Since $a, b$ and $d$ are both positive, the quadratic $aX^2 - bX - d$ must have a positive and a negative root. Therefore, this quadratic is negative if and only if $X$ is less than the positive root; that is, we have

$$aX^2 - bX - d \leq 0$$
$$\iff X \leq \frac{b + \sqrt{b^2 + 4ad}}{2a}$$
$$\implies X \leq \frac{b + \sqrt{ad}}{a}.$$

$\square$

**Proposition E.4.** *Let $\delta_n^q$ respectively be the upper bound on the Rademacher complexity of $\mathcal{M}$. For any $\eta \in (0, 1)$, define $\delta^q := \delta_n^q + c_0^q \sqrt{\frac{\log(c_1^q/\eta)}{n}}$ for some constants $c_0^q, c_1^q$; then under assump. 6.1, we have with probability $1 - \eta$ that*

$$\sup_{m \in \mathcal{M}} \Phi_q^{n,\lambda_m}(q_0, m; \hat{p}) \leq \frac{1}{\lambda_m} \left\| \frac{1}{p} - \frac{1}{\hat{p}} \right\|_2^2 + \left\{ \frac{\lambda_m}{2} + \frac{c^2 C_1^2}{\lambda_m} + cC_1 \right\} (\delta^q)^2$$

*Proof.* To relate $\|m\|_{2,n}^2$ and $\|m\|_2^2$, by Lemma E.1, it holds with probability at least $1 - \eta$ that

$$\left| \|m\|_{2,n}^2 - \|m\|_2^2 \right| \leq \frac{1}{2} \|m\|_2^2 + \frac{1}{2} (\delta^q)^2, \qquad \forall m \in \mathcal{M}, \tag{17}$$

as long as we choose $t$ equal to $\delta^q = \delta_n^q + c_0 \sqrt{\frac{\log(c_1/\eta)}{n}}$, where $\delta_n^q$ is an upper bound on the critical radius of $\mathcal{M}$. On the other hand, to relate $\Phi_q^{n,\lambda_m}(\hat{q}, m; \hat{p}) = \Phi_q^n(\hat{q}, m; \hat{p}) - \lambda_m \|m\|_2^2$ to its empirical version, we apply Lemma E.2 to $\ell(a_1, a_2) := a_1 a_2, a_1 = m(A, W, X), a_2 = q_0(A, Z, X) - \frac{1}{\hat{p}(A|W,X)}$ that is $C_1$-Lipschitz with respect to $a_1$ by noting $q_0(A, Z, X) - \frac{1}{\hat{p}(A|W,X)}$ is in $[-C_1, C_1]$ with some constants $C_1 = 1/\|\hat{p}\|_\infty + \|Q\|_\infty$:

$$|\ell(a_1, a_2) - \ell(a_1', a_2)| \leq C_1 |a_1 - a_1'|.$$

Therefore, we have that there exists a positive constant $c$ such that, with probability at least $1 - \eta$,

$$\left| \Phi_q^n(q_0, m; \hat{p}) - \Phi_q(q_0, m; \hat{p}) \right| \leq cC_1 \left\{ \delta^q \|m\|_2 + (\delta^q)^2 \right\}. \qquad \forall m \in \mathcal{M} \tag{18}$$

Thus, we can further deduce that, for some absolute constants $c > 0$, with probability at least $1 - \eta$,

$$\sup_{m \in \mathcal{M}} \Phi_q^{n,\lambda_m}(q_0, m; \hat{p}) = \sup_{m \in \mathcal{M}} \left\{ \Phi_q^n(q_0, m; \hat{p}) - \lambda_m \|m\|_{2,n}^2 \right\}$$
$$\overset{(1)}{\leq} \sup_{m \in \mathcal{M}} \left\{ \Phi_q(q_0, m; \hat{p}) + cC_1 \{ \delta^q \|m\|_2 + (\delta^q)^2 \} - \lambda_m \|m\|_{2,n}^2 \right\}$$
$$\overset{(2)}{\leq} \sup_{m \in \mathcal{M}} \left\{ \Phi_q(q_0, m; \hat{p}) + cC_1 \{ \delta^q \|m\|_2 + (\delta^q)^2 \} - \frac{\lambda_m}{2} \|m\|_2^2 + \frac{\lambda_m}{2} (\delta^q)^2 \right\},$$

where (1) is derived from Eq. 18, and (2) is derived from Eq. 17. We cam further bound the right-hand side of the above inequality as

$$
\begin{aligned}
\sup_{m\in\mathcal{M}} \Phi_q^{n,\lambda_m}(q_0,m;\hat{p}) \leq{}& \sup_{m\in\mathcal{M}} \left\{ \Phi_q(q_0,m;\hat{p}) - \frac{\lambda_m}{4}\|m\|_2^2 \right\} \\
& + \sup_{m\in\mathcal{M}} \left\{ cC_1\{\delta^q\|m\|_2 + (\delta^q)^2\} - \frac{\lambda_m}{4}\|m\|_2^2 + \frac{\lambda_m}{2}(\delta^q)^2 \right\} \\
\overset{(1)}{\leq}{}& \sup_{m\in\mathcal{M}} \Phi_q(q_0,m;p) + \left\{ \frac{\lambda_m}{2} + \frac{c^2C_1^2}{\lambda_m} + cC_1 \right\}(\delta^q)^2 \\
& + \sup_{m\in\mathcal{M}} \left\{ \Phi_q(q_0,m;\hat{p}) - \Phi_q(q_0,m;p) - \frac{\lambda_m}{4}\|m\|_2^2 \right\} \\
\overset{(2)}{\leq}{}& \sup_{m\in\mathcal{M}} \left\{ \mathbb{E}\left[\left(\frac{1}{p}-\frac{1}{\hat{p}}\right)m\right] - \frac{\lambda_m}{4}\|m\|_2^2 \right\} + \left\{ \frac{\lambda_m}{2} + \frac{c^2C_1^2}{\lambda_m} + cC_1 \right\}(\delta^q)^2 \\
\overset{(3)}{\leq}{}& \sup_{m\in\mathcal{M}} \left\{ \left\|\frac{1}{p}-\frac{1}{\hat{p}}\right\|_2 \|m\|_2 - \frac{\lambda_m}{4}\|m\|_2^2 \right\} + \left\{ \frac{\lambda_m}{2} + \frac{c^2C_1^2}{\lambda_m} + cC_1 \right\}(\delta^q)^2 \\
\overset{(1)}{\leq}{}& \frac{1}{\lambda_m}\left\|\frac{1}{p}-\frac{1}{\hat{p}}\right\|_2^2 + \left\{ \frac{\lambda_m}{2} + \frac{c^2C_1^2}{\lambda_m} + cC_1 \right\}(\delta^q)^2
\end{aligned}
$$

where (1) from the fact $\sup_{\|m\|_2}\{a\|m\|_2 - b\|m\|_2^2\} \leq a^2/4b$ for any $b > 0$, (2) holds from the fact that $\Phi_q(q_0,m;p) = 0$ and (3) holds from Cauchy's inequality.

$\square$

**Theorem 6.2.** *Let $\delta_n^q$ respectively be the upper bound on the Rademacher complexity of $\mathcal{M}$. For any $\eta \in (0,1)$, define $\delta^q := \delta_n^q + c_0^q\sqrt{\frac{\log(c_1^q/\eta)}{n}}$ for some constants $c_0^q, c_1^q$; then under assump. 6.1, we have with probability $1 - \eta$ that*

$$
\left\|\mathrm{proj}_q(\hat{q}-q_0)\right\|_2 = O\left( \delta^q\sqrt{\lambda_m^2 + \lambda_m + 1} + \left\|\frac{1}{p}-\frac{1}{\hat{p}}\right\|_2 \right), \quad p \text{ stands for } p(a|w,x).
$$

*Proof.* First we note that,

$$
\begin{aligned}
\sup_{m\in\mathcal{M}} \Phi_q^{n,\lambda_m}(\hat{q},m;\hat{p}) ={}& \sup_{m\in\mathcal{M}} \left\{ \Phi_q^n(\hat{q},m;\hat{p}) - \Phi_q^n(q_0,m;\hat{p}) + \Phi_q^n(q_0,m;\hat{p}) - \lambda_m\|m\|_{2,n}^2 \right\} \\
\geq{}& \underbrace{\sup_{m\in\mathcal{M}} \left\{ \Phi_q^n(\hat{q},m;\hat{p}) - \Phi_q^n(q_0,m;\hat{p}) - 2\lambda_m\|m\|_{2,n}^2 \right\}}_{(\star)} \\
& + \inf_{m\in\mathcal{M}} \left\{ \Phi_q^n(q_0,m;\hat{p}) + \lambda_m\|m\|_{2,n}^2 \right\}
\end{aligned}
$$

By the symmetry of $\mathcal{M}$, we have

$$
\begin{aligned}
\inf_{m\in\mathcal{M}} \left\{ \Phi_q^n(q_0,m;\hat{p}) + \lambda_m\|m\|_{2,n}^2 \right\} ={}& \inf_{-m\in\mathcal{M}} \left\{ \Phi_q^n(q_0,-m;\hat{p}) + \lambda_m\|m\|_{2,n}^2 \right\} \\
={}& \inf_{-m\in\mathcal{M}} \left\{ -\Phi_q^n(q_0,m;\hat{p}) + \lambda_m\|m\|_{2,n}^2 \right\} \\
={}& -\sup_{-m\in\mathcal{M}} \left\{ \Phi_q^n(q_0,m;\hat{p}) - \lambda_m\|m\|_{2,n}^2 \right\} \\
={}& -\sup_{-m\in\mathcal{M}} \Phi_q^{n,\lambda_m}(q_0,m;\hat{p}) \\
={}& -\sup_{m\in\mathcal{M}} \Phi_q^{n,\lambda_m}(q_0,m;\hat{p})
\end{aligned}
$$

In the sequel, we upper and lower bound term $(\star)$ respectively.

**(i). Upper bound of term $(\star)$.**

By definition of the estimator $\hat{q}$ and the assumption $q_0 \in \mathcal{Q}$, we have

$$\sup_{m \in \mathcal{M}} \Phi_q^{n,\lambda_m}(\hat{q}, m; \hat{p}) \leq \sup_{m \in \mathcal{M}} \Phi_q^{n,\lambda_m}(q_0, m; \hat{p}).$$

We have

$$(\star) \leq \sup_{m \in \mathcal{M}} \Phi_q^{n,\lambda_m}(\hat{q}, m; \hat{p}) + \sup_{m \in \mathcal{M}} \Phi_q^{n,\lambda_m}(q_0, m; \hat{p})$$

$$\leq 2 \sup_{m \in \mathcal{M}} \Phi_q^{n,\lambda_m}(q_0, m; \hat{p}) \leq 2 \left( \frac{1}{\lambda_m} \left\| \frac{1}{p} - \frac{1}{\hat{p}} \right\|_2^2 + \left\{ \frac{\lambda_m}{2} + \frac{c^2 C_1^2}{\lambda_m} + c C_1 \right\} (\delta^q)^2 \right)$$

**(ii). Lower bound of term $(\star)$.**

We now invoke Lemma E.2 with $\ell(a_1, a_2)$, $a_1 = m$ and $a_2 = q - q_0$ that is $C_2$-Lipschitz with respect to $a_1$ by noting $q(A, Z, X) - q_0(A, Z, X)$ is in $[-C_2, C_2]$ with some constants $C_2 = 2\|Q\|_\infty$:

$$|\ell(a_1, a_2) - \ell(a_1', a_2)| \leq C_2 |a_1 - a_1'|.$$

Therefore, we have that there exists a positive constant $c$ such that, with probability at least $1 - \eta$,

$$\left| \left\{ \Phi_q^n(q, m; \hat{p}) - \Phi_q^n(q_0, m; \hat{p}) \right\} - \left\{ \Phi_q(q, m; \hat{p}) - \Phi_q(q_0, m; \hat{p}) \right\} \right| \leq c C_2 \{ \delta^q \|m\|_2 + (\delta^q)^2 \} \tag{19}$$

Now we are ready to prove the lower bound on term $(\star)$. Since $m_q := \frac{1}{2\lambda_m} \mathrm{proj}_q(q - q_0) \in \mathcal{M}$, and Star-shaped, we have $\frac{m_q}{2} \in \mathcal{M}$

$$(\star) = \sup_{m \in \mathcal{M}} \left\{ \Phi_q^n(\hat{q}, m; \hat{p}) - \Phi_q^n(q_0, m; \hat{p}) - 2\lambda_m \|m\|_{2,n}^2 \right\}$$

$$\geq \Phi_q^n\left(\hat{q}, \frac{m_q}{2}; \hat{p}\right) - \Phi_q^n\left(q_0, \frac{m_q}{2}; \hat{p}\right) - \frac{\lambda_m}{2} \|m_q\|_{2,n}^2$$

$$\geq \underbrace{\Phi_q\left(\hat{q}, \frac{m_q}{2}; \hat{p}\right) - \Phi_q\left(q_0, \frac{m_q}{2}; \hat{p}\right)}_{(\diamond)} - c C_2 \left\{ \delta^q \left\| \frac{m_q}{2} \right\|_2 + (\delta^q)^2 \right\} - \frac{\lambda_m}{2} \left( \frac{3}{2} \|m_q\|_2^2 + \frac{(\delta^q)^2}{2} \right)$$

where the last line holds from the Eq. 17 and 19. For the term $(\diamond)$, we have

$$(\diamond) = \Phi_q\left(\hat{q}, \frac{m_q}{2}; p\right) + \Phi_q\left(\hat{q}, \frac{m_q}{2}; \hat{p}\right) - \Phi_q\left(\hat{q}, \frac{m_q}{2}; p\right) - \Phi_q\left(q_0, \frac{m_q}{2}; p\right)$$

$$+ \Phi_q\left(q_0, \frac{m_q}{2}; p\right) - \Phi_q\left(q_0, \frac{m_q}{2}; \hat{p}\right)$$

$$= \Phi_q\left(\hat{q}, \frac{m_q}{2}; p\right) + \mathbb{E}\left[\left(\frac{1}{p} - \frac{1}{\hat{p}}\right) \frac{m_q}{2}\right] + \mathbb{E}\left[\left(\frac{1}{\hat{p}} - \frac{1}{p}\right) \frac{m_q}{2}\right] = \Phi_q\left(\hat{q}, \frac{m_q}{2}; p\right)$$

where we used $\Phi_q\left(q_0, \frac{m_q}{2}; p\right) = 0$. Moreover, since we have

$$\mathbb{E}\left[\hat{q}(A, Z, X) - \frac{1}{p(A \mid W, X)} \mid A, W, X\right] = \mathbb{E}[\hat{q}(A, Z, X) - q_0(A, Z, X) \mid A, W, X]$$

$$= \mathrm{proj}_q(\hat{q} - q_0)$$

and $m_q = \frac{1}{2\lambda_m} \mathrm{proj}_q(\hat{q} - q_0)$, we have

$$\Phi_q\left(\hat{q}, \frac{m_q}{2}; p\right) = \frac{1}{2} \mathbb{E}\left[\left(q(Z, A, X) - \frac{1}{p(A \mid W, X)}\right) m_q(A, W, X)\right]$$

$$= \frac{1}{2} \mathbb{E}\left[m_q(A, W, X) \mathbb{E}\left[q(A, Z, X) - \frac{1}{p(A \mid W, X)} \mid A, W, X\right]\right]$$

$$= \frac{1}{4\lambda_m} \mathbb{E}\left[\left(\mathrm{proj}_q(\hat{q} - q_0)\right)^2\right] = \lambda_m \|m_q\|_2^2$$

Therefore, we obtain the lower bound of $(\star)$:

$$(\star) \geq \lambda_m \|m_q\|_2^2 - cC_2 \left\{ \delta^q \left\| \frac{m_q}{2} \right\|_2 + (\delta^q)^2 \right\} - \frac{\lambda_m}{2} \left( \frac{3}{2} \|m_q\|_2^2 + \frac{(\delta^q)^2}{2} \right)$$

**(iii). Combining upper bound and lower bound of term $(\star)$.**

Now we are ready to combine the upper bound and lower bound of $(\star)$.

$$\lambda_m \|m_q\|_2^2 - cC_2 \left\{ \delta^q \left\| \frac{m_q}{2} \right\|_2 + (\delta^q)^2 \right\} - \frac{\lambda_m}{2} \left( \frac{3}{2} \|m_q\|_2^2 + \frac{(\delta^q)^2}{2} \right)$$
$$\leq 2 \left( \frac{1}{\lambda_m} \left\| \frac{1}{p} - \frac{1}{\hat{p}} \right\|_2^2 + \left\{ \frac{\lambda_m}{2} + \frac{c^2 C_1^2}{\lambda_m} + cC_1 \right\} (\delta^q)^2 \right)$$

This give a quadratic inequality on $\|m_q\|_2$, i.e.

$$\lambda_m \|m_q\|_2^2 - \underbrace{2cC_2\delta^q}_{(B)} \|m_q\|_2 - \underbrace{\left( \frac{8}{\lambda_m} \left\| \frac{1}{p} - \frac{1}{\hat{p}} \right\|_2^2 + \left\{ 5\lambda_m + \frac{8c^2 C_1^2}{\lambda_m} + 8cC_1 + 4cC_2 \right\} (\delta^q)^2 \right)}_{(C)} \leq 0$$

By Lemma E.3, we have that

$$\|m_q\|_2 \leq \frac{B + \sqrt{B^2 + 4\lambda_m C}}{2\lambda_m} \leq \frac{1}{\lambda_m} \left( B + \sqrt{\lambda_m C} \right)$$

Applying the definition of A and B, we conclude that, with probability at least $1 - \eta$,

$$\|m_q\|_2 \leq \frac{2cC_2\delta^q}{\lambda_m} + \frac{1}{\lambda_m} \sqrt{\lambda_m \left( \frac{8}{\lambda_m} \left\| \frac{1}{p} - \frac{1}{\hat{p}} \right\|_2^2 + \left\{ 5\lambda_m + \frac{8c^2 C_1^2}{\lambda_m} + 8cC_1 + 4cC_2 \right\} (\delta^q)^2 \right)}$$
$$\leq \left( \frac{2cC_2}{\lambda_m} + \sqrt{5 + \frac{8cC_1 + 4cC_2}{\lambda_m} + \frac{8c^2 C_1^2}{\lambda_m^2}} \right) \delta^q + \frac{2\sqrt{2}}{\lambda_m} \left\| \frac{1}{p} - \frac{1}{\hat{p}} \right\|_2$$
$$\leq \sqrt{2 \left( 5 + \frac{8cC_1 + 4cC_2}{\lambda_m} + \frac{4c^2 (2C_1^2 + C_2^2)}{\lambda_m^2} \right)} \delta^q + \frac{2\sqrt{2}}{\lambda_m} \left\| \frac{1}{p} - \frac{1}{\hat{p}} \right\|_2$$
$$\lesssim \sqrt{1 + \frac{1}{\lambda_m} + \frac{1}{\lambda_m^2}} \delta^q + \frac{1}{\lambda_m} \left\| \frac{1}{p} - \frac{1}{\hat{p}} \right\|_2$$

where the third line holds from the $\sqrt{a} + \sqrt{b} \leq \sqrt{2(a+b)}$. According Eq. 11, we have

$$\sqrt{\mathcal{L}_q(q; p)} = \sqrt{\mathbb{E}\left[ \left( \text{proj}_q (\hat{q} - q_0) \right)^2 \right]} = \sqrt{4\lambda_m^2 \mathbb{E}\left[ (m_q)^2 \right]} = 2\lambda_m \|m_q\|_2,$$

we can bound the Projected RMSE by

$$\left\| \text{proj}_q(\hat{q} - q_0) \right\|_2 = \sqrt{\mathcal{L}_q(q; p)} = 2\lambda_m \|m_q\|_2 \lesssim \delta^q \sqrt{\lambda_m^2 + \lambda_m + 1} + \left\| \frac{1}{p} - \frac{1}{\hat{p}} \right\|_2$$

$$\square$$

For bridge function $h$, we also the similar theorem. We first give some assumption.

**Assumption E.5.** (1) (Boundness) $\|\mathcal{G}\|_\infty < \infty$ and $y$ is uniformly bounded; (2) (Symmetric) $\mathcal{G}$ is a symmetric class, i.e, if $g \in \mathcal{G}$, then $-g \in \mathcal{G}$; (3) (Star-shaped) $\mathcal{G}$ is star-shaped class, that is for each function $g$ in the class, $\alpha g$ for any $\alpha \in [0, 1]$ also belongs to the class; (4) (Realizability) $h_0 \in \mathcal{H}$; (5) (Closedness) $\frac{1}{2\lambda_g} \text{proj}_h(h - h_0) \in \mathcal{H}$.

**Theorem E.6.** *Let $\delta_n^h$ respectively be the upper bound on the Rademacher complexity of $\mathcal{G}$. For any $\eta \in (0,1)$, define $\delta^h := \delta_n^h + c_0^h \sqrt{\frac{\log(c_1^h/\eta)}{n}}$ for some constants $c_0^h, c_1^h$; then under assump. E.5, we have with probability $1 - \eta$ that*

$$\left\| \mathrm{proj}_h(\hat{h} - h_0) \right\|_2 = O\left( \delta^h \sqrt{\lambda_g^2 + \lambda_g + 1} \right)$$

The proof of the Thm. E.6 is detailed in Kallus et al. (2021).

### E.4 PROOF OF THEOREM 6.4

We prove the bias and variance of the PKDR estimator given Eq. 7 respectively. The proof method comes from Colangelo & Lee (2020); Kallus & Uehara (2020).

**Theorem E.7.** *Under assump. 3.1-3.4 and 4.1, suppose $\|\hat{h} - h\|_2 = o(1)$, $\|\hat{q} - q\|_2 = o(1)$ and $\|\hat{h} - h\|_2 \|\hat{q} - q\|_2 = o((nh_{\mathrm{bw}})^{-1/2})$, $nh_{\mathrm{bw}}^5 = O(1)$, $nh_{\mathrm{bw}} \to \infty$, $h_0(a,w,x), p(a,z|w,x)$ and $p(a,w|z,x)$ are twice continuously differentiable wrt $a$, as well as $h_0, q_0, \hat{h}, \hat{q}$ are uniformly bounded. Then for any $a$, we have the following for the bias of the PKDR estimator given Eq. 7:*

$$\mathbb{E}\left[ \hat{\beta}(a) \right] - \beta(a) = \frac{h_{\mathrm{bw}}^2}{2} \kappa_2(K)\mathrm{B} + o((nh_{\mathrm{bw}})^{-1/2}),$$

*where $B = \mathbb{E}[q_0(a,Z,X)[2\frac{\partial}{\partial A}h_0(a,W,X)\frac{\partial}{\partial A}p(a,W \mid Z,X) + \frac{\partial^2}{\partial A^2}h_0(a,W,X)]]$.*

*Proof.* We calculate the expectation of the estimator for a single data point. For simplicity, we treat this data point as a random variable. We have

$$
\begin{aligned}
\mathbb{E}\left[ \hat{\beta}(a) - \beta(a) \right] &= \mathbb{E}\left[ \hat{\beta}(a) \right] - \beta(a) \\
&= \underbrace{\mathbb{E}\left[ K_{h_{\mathrm{bw}}}(A-a) \left\{ \left(Y - \hat{h}(a,W,X)\right) \hat{q}(a,Z,X) \right\} \right]}_{\textbf{(I)}} \\
&\quad - \underbrace{\mathbb{E}\left[ K_{h_{\mathrm{bw}}}(A-a) \left\{ (Y - h_0(a,W,X)) q_0(a,Z,X) \right\} \right]}_{\textbf{(II)}} \\
&\quad + \underbrace{\mathbb{E}\left[ \hat{h}(a,W,X) - h_0(a,W,X) \right]}_{\textbf{(III)}} \\
&\quad + \underbrace{\mathbb{E}\left[ K_{h_{\mathrm{bw}}}(A-a)(Y - h_0(a,W,X)) q_0(a,Z,X) \right]}_{\textbf{(IV)}}
\end{aligned}
$$

For the **(IV)** term, we first have

$$
\begin{aligned}
&\mathbb{E}\left[ K_{h_{\mathrm{bw}}}(A-a) q_0(a,Z,X)(Y - h_0(a,W,X)) \right] \\
=&\mathbb{E}\left[ K_{h_{\mathrm{bw}}}(A-a) q_0(a,Z,X) \mathbb{E}\left[ (Y - h_0(a,W,X)) \mid A,Z,X \right] \right] \\
=&\mathbb{E}\left[ K_{h_{\mathrm{bw}}}(A-a) q_0(a,Z,X) \mathbb{E}\left[ (h_0(A,W,X) - h_0(a,W,X)) \mid A,Z,X \right] \right] \\
=&\mathbb{E}\left[ q_0(a,Z,X) \mathbb{E}\left[ K_{h_{\mathrm{bw}}}(A-a)(h_0(A,W,X) - h_0(a,W,X)) \mid Z,X \right] \right] \\
=&\mathbb{E}\left[ q_0(a,Z,X) \int K_{h_{\mathrm{bw}}}(a'-a)(h_0(a',w,x) - h_0(a,w,x)) p(a',w \mid z,x) \, \mathrm{d}\mu(a',w) \right] \\
=&\mathbb{E}\left[ q_0(a,Z,X) \int K(u)(h_0(a+h_{\mathrm{bw}}u,w,x) - h_0(a,w,x)) p(a+h_{\mathrm{bw}}u,w \mid z,x) \, \mathrm{d}\mu(u,w) \right]
\end{aligned}
$$

where the last line holds from $a' = h_{\mathrm{bw}} u + a$. Consider Taylor expansion of $h_0(a, w, x)$ and $p(a, w \mid z, x)$ around $A = a$:

$$p(h_{\mathrm{bw}} u + a, w \mid z, x) - p(a, w \mid z, x) = h_{\mathrm{bw}} u \frac{\partial}{\partial A} p(a, w \mid z, x) + O(h_{\mathrm{bw}}^2)$$

$$h_0(a + h_{\mathrm{bw}} u, w, x) - h_0(a, w, x) = h_{\mathrm{bw}} u \left( \frac{\partial}{\partial A} h_0(a, w, x) \right)$$
$$+ \frac{(h_{\mathrm{bw}} u)^2}{2} \left( \frac{\partial^2}{\partial A^2} h_0(a, w, x) \right) + O(h_{\mathrm{bw}}^3)$$

Then, we can compute the conditional expectation by integrating the approximation to the density term by term. Here, $\kappa_j(K)$ represents the jth kernel moment, defined as $\kappa_j(K) = \int u^j K(u) du$. It's important to note that for a symmetric kernel, the odd-order moments integrate to 0. Therefore, we have

$$\mathbb{E}\left[ K_{h_{\mathrm{bw}}}(A - a) q_0(a, Z, X)(Y - h_0(a, W, X)) \right]$$

$$= \mathbb{E}\left[ q_0(a, Z, X) \int K(u)(h_0(a + h_{\mathrm{bw}} u, w, x) - h_0(a, w, x)) p(a + h_{\mathrm{bw}} u, w \mid z, x) \,\mathrm{d}\mu(u, w) \right]$$

$$= h_{\mathrm{bw}}^2 \kappa_2(K) \mathbb{E}\left[ q_0(a, Z, X) \left[ \frac{\partial}{\partial A} h_0(a, W, X) \frac{\partial}{\partial A} p(a, W \mid Z, X) + \frac{1}{2} \left( \frac{\partial^2}{\partial A^2} h_0(a, W, X) \right) \right] \right]$$
$$+ o(h_{\mathrm{bw}}^2)$$

For the **(I)-(III)** term, we have

**(I) $-$ (II) $+$ (III)**

$$= \mathbb{E}\left[ K_{h_{\mathrm{bw}}}(A - a)(\hat{q}(a, Z, X) - q_0(a, Z, X)) \left( h_0(a, W, X) - \hat{h}(a, W, X) \right) \right] \tag{20}$$

$$+ \mathbb{E}\left[ K_{h_{\mathrm{bw}}}(A - a)(\hat{q}(a, Z, X) - q_0(a, Z, X))(Y - h_0(a, W, X)) \right] \tag{21}$$

$$+ \mathbb{E}\left[ K_{h_{\mathrm{bw}}}(A - a) q_0(a, Z, X) \left( h_0(a, W, X) - \hat{h}(a, W, X) \right) - \left( h_0(a, W, X) - \hat{h}(a, W, X) \right) \right]. \tag{22}$$

We will explain in turn that the above three convergence rates are $o((n h_{\mathrm{bw}})^{-1/2})$, $o(1) \times O(h_{\mathrm{bw}}^2)$ and $o(1) \times O(h_{\mathrm{bw}}^2)$ respectively. From now on, we prove Eq. 21 is $o(1) \times O(h_{\mathrm{bw}}^2)$.

$$\mathbb{E}\left[ K_{h_{\mathrm{bw}}}(A - a)(\hat{q}(a, Z, X) - q_0(a, Z, X))(Y - h_0(a, W, X)) \right]$$
$$= \mathbb{E}\left[ K_{h_{\mathrm{bw}}}(A - a)(\hat{q}(a, Z, X) - q_0(a, Z, X)) \mathbb{E}\left[ (Y - h_0(a, W, X)) \mid A, Z, X \right] \right]$$
$$\overset{(1)}{=} \mathbb{E}\left[ K_{h_{\mathrm{bw}}}(A - a)(\hat{q}(a, Z, X) - q_0(a, Z, X)) \mathbb{E}\left[ (h_0(A, W, X) - h_0(a, W, X)) \mid A, Z, X \right] \right]$$
$$= \mathbb{E}\left[ K_{h_{\mathrm{bw}}}(A - a)(\hat{q}(a, Z, X) - q_0(a, Z, X))(h_0(A, W, X) - h_0(a, W, X)) \right]$$
$$= \mathbb{E}\left[ (\hat{q}(a, Z, X) - q_0(a, Z, X)) \mathbb{E}\left[ K_{h_{\mathrm{bw}}}(A - a)(h_0(A, W, X) - h_0(a, W, X)) \mid Z, X \right] \right]$$
$$\overset{(2)}{=} \mathbb{E}\left[ (\hat{q}(a, Z, X) - q_0(a, Z, X)) \{ O(h_{\mathrm{bw}}^2) \} \right]$$
$$\overset{(3)}{=} o(1) \times O(h_{\mathrm{bw}}^2),$$

where (1) is derived from Eq. 2, (3) is derived from assumption $\|\hat{q} - q\|_2 = o(1)$ and (2) is because

$$\mathbb{E}\left[ K_{h_{\mathrm{bw}}}(A - a)(h_0(A, W, X) - h_0(a, W, X)) \mid Z, X \right]$$
$$= \int K_{h_{\mathrm{bw}}}(a' - a)(h_0(a', w, x) - h_0(a, w, x)) p(a', w \mid z, x) \,\mathrm{d}\mu(a', w)$$
$$= \int K(u)(h_0(a + h_{\mathrm{bw}} u, w, x) - h_0(a, w, x)) p(a + h_{\mathrm{bw}} u, w \mid z, x) \,\mathrm{d}\mu(u, w)$$
$$= \int K(u) \left( h_{\mathrm{bw}} u \frac{\partial}{\partial A} h_0(a, w, x) + O(h_{\mathrm{bw}}^2) \right)(p(a, w \mid z, x) + O(h_{\mathrm{bw}} u)) \,\mathrm{d}\mu(u, w)$$
$$= O(h_{\mathrm{bw}}^2).$$

Next, we prove Eq. 22 is $o(1) \times O(h_{\mathrm{bw}}^2)$.

$$\mathbb{E}[K_{h_{\mathrm{bw}}}(A - a) q_0(a, Z, X) (h_0(a, W, X) - \hat{h}(a, W, X)) - (h_0(a, W, X) - \hat{h}(a, W, X))]$$
$$= \mathbb{E}[(h_0(a, W, X) - \hat{h}(a, W, X)) \mathbb{E}[K_{h_{\mathrm{bw}}}(A - a) q_0(a, Z, X) - 1 \mid W, X]].$$

We consider

$$\mathbb{E}[K_{h_{\mathrm{bw}}}(A - a) q_0(a, Z, X) - 1 \mid W, X]$$

$$= \int (K_{h_{\mathrm{bw}}}(a' - a) q_0(a, z, x) - 1) p(a', z \mid w, x) \,\mathrm{d}\mu(a', z)$$

$$= \int K_{h_{\mathrm{bw}}}(a' - a) q_0(a, z, x) p(a', z \mid w, x) \,\mathrm{d}\mu(a', z) - 1$$

$$= \int K(u) q_0(a, z, x) p(a + h_{\mathrm{bw}} u, z \mid w, x) \,\mathrm{d}\mu(u, z) - 1$$

$$= \int K(u) q_0(a, z, x) \left( p(a, z \mid w, x) + h_{\mathrm{bw}} u \frac{\partial}{\partial A} h_0(a, w, x) + O\left(h_{\mathrm{bw}}^2\right) \right) \mathrm{d}\mu(u, z) - 1$$

$$= \int q_0(a, z, x) p(a, z \mid w, x) \,d\mu(z) - 1 + O\left(h_{\mathrm{bw}}^2\right),$$

where we use the first-order Taylor expansion of $p(a, z \mid w, x)$:

$$p(h_{\mathrm{bw}} u + a, z \mid w, x) = p(a, z \mid w, x) + h_{\mathrm{bw}} u \frac{\partial}{\partial A} h_0(a, w, x) + O\left(h_{\mathrm{bw}}^2\right).$$

Therefore, we have

$$\mathbb{E}\left[ \left( h_0(a, W, X) - \hat{h}(a, W, X) \right) \left( \int q_0(a, z, x) p(a, z \mid w, x) \,\mathrm{d}\mu(z) - 1 + O\left(h_{\mathrm{bw}}^2\right) \right) \right]$$

$$\overset{(1)}{=} \mathbb{E}\left[ \left( h_0(a, W, X) - \hat{h}(a, W, X) \right) \left( \int q_0(a, z, x) p(a, z \mid w, x) \,\mathrm{d}\mu(z) - 1 \right) \right]$$
$$+ o(1) \times O\left(h_{\mathrm{bw}}^2\right)$$

where (1) is derived from assumption $\|\hat{h} - h\|_2 = o(1)$. We assert that the first expression is 0:

$$\mathbb{E}\left[ \left( h_0(a, W, X) - \hat{h}(a, W, X) \right) \left( \int q_0(a, z, x) p(a, z \mid w, x) \,\mathrm{d}\mu(z) - 1 \right) \right]$$

$$= \int \left( h_0(a, w, x) - \hat{h}(a, w, x) \right) q_0(a, z, x) p(a, z, w, x) \,\mathrm{d}\mu(z, w, x)$$

$$- \int \left( h_0(a, w, x) - \hat{h}(a, w, x) \right) p(w, x) \,\mathrm{d}\mu(w, x)$$

$$\overset{(1)}{=} \int \hat{h}(a, w, x) p(w, x) \,\mathrm{d}\mu(w, x) - \int \hat{h}(a, w, x) q_0(a, z, x) p(a, z, w, x) \,\mathrm{d}\mu(z, w, x)$$

$$= \int \hat{h}(a, w, x) p(w, x) \,\mathrm{d}\mu(w, x) - \int \hat{h}(a, w, x) \frac{p(a, w, x)}{p(a \mid w, x)} \mathrm{d}\mu(w, x) = 0$$

where we used the following property for (1)

$$\beta(a) = \int h_0(a, w, x) p(w, x) \,\mathrm{d}\mu(w, x)$$

$$= \int h_0(a, w, x) q_0(a, z, x) p(a, z, w, x) \,\mathrm{d}\mu(z, w, x).$$

By assumption $\|\hat{h} - h\|_2 \|\hat{q} - q\|_2 = o((nh_{\mathrm{bw}})^{-1/2})$, we have Eq. 20 is $o((nh_{\mathrm{bw}})^{-1/2})$. Combining these terms we get

$$\mathbf{(I)} - \mathbf{(II)} + \mathbf{(III)} = o((nh_{\mathrm{bw}})^{-1/2}) + o(1) \times O(h_{\mathrm{bw}}^2) + o(1) \times O(h_{\mathrm{bw}}^2) = o((nh_{\mathrm{bw}})^{-1/2})$$

where we use $nh_{\mathrm{bw}}^5 = O(1)$. Therefore,

$$\mathbb{E}\left[ \hat{\beta}(a) \right] - \beta(a) = \frac{h_{\mathrm{bw}}^2}{2} \kappa_2(K) B + o((nh_{\mathrm{bw}})^{-1/2}),$$

where $B = \mathbb{E}[q_0(a, Z, X)[2\frac{\partial}{\partial A} h_0(a, W, X) \frac{\partial}{\partial A} p(a, W \mid Z, X) + \frac{\partial^2}{\partial A^2} h_0(a, W, X)]]$. $\qquad\square$

**Theorem E.8.** *Under assump. 3.1-3.4 and 4.1, suppose $\|\hat{h} - h\|_2 = o(1)$, $\|\hat{q} - q\|_2 = o(1)$ and $\|\hat{h} - h\|_2\|\hat{q} - q\|_2 = o((nh_{\mathrm{bw}})^{-1/2})$, $nh_{\mathrm{bw}}^5 = O(1)$, $nh_{\mathrm{bw}} \to \infty$, $h_0(a, w, x), p(a, z|w, x)$ and $p(a, w|z, x)$ are twice continuously differentiable wrt $a$ as well as $h_0, q_0, \hat{h}, \hat{q}$ are uniformly bounded. Then for any $a$, we have the following for the variance of the PKDR estimator given Eq. 7:*

$$\mathrm{Var}[\hat{\beta}(a)] = \frac{\Omega_2(K)}{nh_{\mathrm{bw}}}(V + o(1)),$$

*where $V = \mathbb{E}[\mathbb{I}(A = a)q_0(a, Z, X)^2(Y - h_0(a, W, X))^2]$.*

*Proof.* For convenience, we let

$$m(o; h, q) = K_{h_{\mathrm{bw}}}(A - a)(Y - h(a, W, X))q(a, Z, X) \tag{23}$$
$$\phi(o; h, q) = m(o; h, q) + h(a, W, X) \tag{24}$$

We first use cross-fitting which allows us to exchange the order of summation and variance. More specifically, we split the data randomly into two halves $O_1$ and $O_2$. Then we have

$$\mathbb{E}\left[\mathbb{E}_n\left[\phi_1\left(o; \hat{h}, \hat{q}\right)\right]^2\right] = \mathbb{E}\left[\mathbb{E}\left[\mathbb{E}_n\left[\phi_1\left(o; \hat{h}, \hat{q}\right)\right]^2 |O_2\right]\right]$$
$$= n^{-1}\mathbb{E}\left[\mathbb{E}\left[\left(\phi_1\left(o; \hat{h}, \hat{q}\right)^2\right)|O_2\right]\right]$$

We will omit this step later, please identify it according to the context. According to the definition of variance, we have

$$\mathrm{Var}\left(\mathbb{E}_n\left[\phi\left(o; \hat{h}, \hat{q}\right)\right] - \beta(a)\right)$$
$$\leq \underbrace{\mathrm{Var}(\mathbb{E}_n[\phi(o; h_0, q_0)] - \beta(a))}_{\textbf{(I)}} + \underbrace{\mathrm{Var}\left(\mathbb{E}_n\left[\phi(o; h_0, q_0) - \phi\left(o; \hat{h}, \hat{q}\right)\right]\right)}_{\textbf{(II)}}$$
$$+ 2\underbrace{\sqrt{\mathrm{Var}(\mathbb{E}_n[\phi(o; h_0, q_0)] - \beta(a))\,\mathrm{Var}\left(\mathbb{E}_n\left[\phi(o; h_0, q_0) - \phi\left(o; \hat{h}, \hat{q}\right)\right]\right)}}_{\textbf{(III)}}$$

We use cross-fitting and by Eq. 24:

$$\textbf{(I)} = \frac{1}{n}\mathrm{Var}\left(m(o; h_0, q_0) + h_0(a, W, X) - \beta(a)\right) \tag{25}$$

We first consider $\mathrm{Var}(m(o; h_0, q_0))$. Since

$$\mathrm{Var}(m(o; h_0, q_0)) = \frac{1}{n}\left(\mathbb{E}\left[m(o; h_0, q_0)^2\right] - (\mathbb{E}[m(o; h_0, q_0)])^2\right)$$
$$\leq \frac{1}{n}\mathbb{E}\left[m(o; h_0, q_0)^2\right],$$

we only consider the second moment of a term in the estimator

$$
\begin{aligned}
&\mathbb{E}\left[m\left(o; h_0, q_0\right)^2\right]\\
=&\mathbb{E}\left[\left(K_{h_{\mathrm{bw}}}\left(A-a\right)\left(Y-h_0\left(a, W, X\right)\right) q_0\left(a, Z, X\right)\right)^2\right]\\
=&\mathbb{E}\left[q_0\left(a, Z, X\right)^2 \mathbb{E}\left[K_{h_{\mathrm{bw}}}\left(A-a\right)^2\left(Y-h_0\left(a, W, X\right)\right)^2 \mid Z, X\right]\right]\\
=&\mathbb{E}\left[q_0\left(a, Z, X\right)^2 \int \frac{1}{h_{\mathrm{bw}}^2} K\left(\frac{a'-a}{h_{\mathrm{bw}}}\right)^2\left(y-h_0\left(a, w, x\right)\right)^2 p\left(a', y, w \mid z, x\right) \mathrm{d}\mu\left(a', y, w\right)\right]\\
=&\mathbb{E}\left[q_0\left(a, Z, X\right)^2 \int \frac{1}{h_{\mathrm{bw}}} K\left(u\right)^2\left(y-h_0\left(a, w, x\right)\right)^2 p\left(a+u h_{\mathrm{bw}}, y, w \mid z, x\right) \mathrm{d}\mu\left(u, y, w\right)\right]\\
=&\mathbb{E}\left[q_0\left(a, Z, X\right)^2 \int \frac{1}{h_{\mathrm{bw}}} K\left(u\right)^2\left(y-h_0\left(a, w, x\right)\right)^2\left(p\left(a, y, w \mid z, x\right)+o\left(h_{\mathrm{bw}}\right)\right) \mathrm{d}\mu\left(u, y, w\right)\right]\\
=&\frac{1}{h_{\mathrm{bw}}}\left\{\Omega_2(K) V+o\left(h_{\mathrm{bw}}\right)\right\}
\end{aligned}
$$

(26)

where $V=\mathbb{E}\left[\mathbb{I}\left(A=a\right) q_0\left(a, Z, X\right)^2\left(Y-h_0\left(a, W, X\right)\right)^2\right]$

Because $h_0$ is bound, we have $\mathrm{Var}\left(h_0\left(a, W, X\right)-\beta\left(a\right)\right)$ is bound. Therefore, substituting the above equation to Eq. 25, we have

$$
\begin{aligned}
\textbf{(I)}=&\frac{1}{n} \mathrm{Var}\left(m\left(o; h_0, q_0\right)+h_0\left(a, W, X\right)-\beta\left(a\right)\right)\\
\leq&\frac{1}{n} \mathrm{Var}\left(m\left(o; h_0, q_0\right)\right)+\frac{1}{n} \mathrm{Var}\left(h_0\left(a, W, X\right)-\beta\left(a\right)\right)\\
&+\frac{2}{n} \sqrt{\mathrm{Var}\left(m\left(o; h_0, q_0\right)\right) \mathrm{Var}\left(h_0\left(a, W, X\right)-\beta\left(a\right)\right)}\\
\leq&\frac{1}{n h_{\mathrm{bw}}}\left\{\Omega_2(K) V+o\left(h_{\mathrm{bw}}\right)\right\}+\frac{C}{n}+\frac{2}{n}\left\{\frac{1}{n h_{\mathrm{bw}}}\left\{V+o\left(h_{\mathrm{bw}}\right)\right\}\right\}^{1/2}\left(\frac{C}{n}\right)^{1/2}\\
\approx&\frac{1}{n h_{\mathrm{bw}}}\left\{\Omega_2(K) V+o\left(h_{\mathrm{bw}}\right)\right\}
\end{aligned}
$$

For the term of **(II)**, we have

$$
\begin{aligned}
\textbf{(II)}=&\mathrm{Var}\left(\mathbb{E}_n\left[\phi\left(o; h_0, q_0\right)-\phi\left(o; \hat{h}, \hat{q}\right)\right]\right)\\
=&\frac{1}{n}\left(\mathbb{E}\left[\left(\phi\left(o; h_0, q_0\right)-\phi\left(o; \hat{h}, \hat{q}\right)\right)^2\right]-\left(\mathbb{E}\left[\phi\left(o; h_0, q_0\right)-\phi\left(o; \hat{h}, \hat{q}\right)\right]\right)^2\right)\\
\leq&\frac{1}{n} \mathbb{E}\left[\left(\phi\left(o; h_0, q_0\right)-\phi\left(o; \hat{h}, \hat{q}\right)\right)^2\right].
\end{aligned}
$$

(27)

Similar to Kallus & Uehara (2020), according to the definition of Eq. 24 and decomposition of Eq. 24 (Eq. 20-22), we have

$$\mathbb{E}\left[\left(\phi\left(o;h_0,q_0\right)-\phi\left(o;\hat{h},\hat{q}\right)\right)^2\right]$$

$$\stackrel{(1)}{=}\mathbb{E}\left[K_{h_{\text{bw}}}\left(A-a\right)^2\left(\hat{q}\left(a,Z,X\right)-q_0\left(a,Z,X\right)\right)^2\left(h_0\left(a,W,X\right)-\hat{h}\left(a,W,X\right)\right)^2\right]$$

$$+\mathbb{E}\left[K_{h_{\text{bw}}}\left(A-a\right)^2\left(\hat{q}\left(a,Z,X\right)-q_0\left(a,Z,X\right)\right)^2\left(Y-h_0\left(a,W,X\right)\right)^2\right]$$

$$+\mathbb{E}\left[K_{h_{\text{bw}}}\left(A-a\right)^2 q_0\left(a,Z,X\right)^2\left(h_0\left(a,W,X\right)-\hat{h}\left(a,W,X\right)\right)^2\right]$$

$$+\mathbb{E}\left[\left(\hat{h}\left(a,W,X\right)-h_0\left(a,W,X\right)\right)^2\right]+\Delta$$

$$\stackrel{(2)}{\lesssim}\mathbb{E}\left[K_{h_{\text{bw}}}\left(A-a\right)^2\left(\hat{q}\left(a,Z,X\right)-q_0\left(a,Z,X\right)\right)^2\left(h_0\left(a,W,X\right)-\hat{h}\left(a,W,X\right)\right)^2\right]$$

$$+\mathbb{E}\left[K_{h_{\text{bw}}}\left(A-a\right)^2\left(\hat{q}\left(a,Z,X\right)-q_0\left(a,Z,X\right)\right)^2\left(Y-h_0\left(a,W,X\right)\right)^2\right]$$

$$+\mathbb{E}\left[K_{h_{\text{bw}}}\left(A-a\right)^2 q_0\left(a,Z,X\right)^2\left(h_0\left(a,W,X\right)-\hat{h}\left(a,W,X\right)\right)^2\right]$$

$$+\mathbb{E}\left[\left(\hat{h}\left(a,W,X\right)-h_0\left(a,W,X\right)\right)^2\right]+O(1)$$

$$\stackrel{(3)}{\lesssim}h_{\text{bw}}^{-1}\max\left\{\mathbb{E}\left[\left\|h_0\left(a,W,X\right)-\hat{h}\left(a,W,X\right)\right\|_2^2\right],\mathbb{E}\left[\left\|\hat{q}\left(a,Z,X\right)-q_0\left(a,Z,X\right)\right\|_2^2\right]\right\}+O(1)$$

$$=o(h_{\text{bw}}^{-1})$$

where (1) is the square expansion of Eq. 20-22 and $\Delta$ is sum of cross terms, (2) is derived from $\hat{h},\hat{q},h,q$ is uniformly bounded, (3) uses the same approach as Eq. 26.

Therefore, substituting the above equation to Eq. 27, we have

$$\textbf{(II)}=\text{Var}\left(\mathbb{E}_n\left[\phi\left(o;h_0,q_0\right)-\phi\left(o;\hat{h},\hat{q}\right)\right]\right)\leq o\left(\left(nh_{\text{bw}}\right)^{-1}\right)\tag{28}$$

For the term of **(III)**, we only need to substitute **(I)** and **(II)** to **(III)**

$$\textbf{(III)}=\sqrt{\text{Var}\left(\mathbb{E}_n\left[\phi\left(o;h_0,q_0\right)\right]-\beta\left(a\right)\right)\text{Var}\left(\mathbb{E}_n\left[\phi\left(o;h_0,q_0\right)-\phi\left(o;\hat{h},\hat{q}\right)\right]\right)}$$

$$=\left\{\frac{1}{nh_{\text{bw}}}\left\{V+o\left(h_{\text{bw}}\right)\right\}\right\}^{1/2}o\left(n^{-1/2}h_{\text{bw}}^{-1/2}\right)$$

Therefore, combining the three terms **(I)**, **(II)** and **(III)**, we get

$$\text{Var}\left(\mathbb{E}_n\left[\phi\left(o;\hat{h},\hat{q}\right)\right]-\beta\left(a\right)\right)$$

$$=\frac{1}{nh_{\text{bw}}}\left\{\Omega_2\left(k\right)V+o\left(h_{\text{bw}}\right)\right\}+2\left\{\frac{1}{nh_{\text{bw}}}\left\{V+o\left(h_{\text{bw}}\right)\right\}\right\}^{1/2}o\left(n^{-1/2}h_{\text{bw}}^{-1/2}\right)+o\left(n^{-1}h_{\text{bw}}^{-1}\right)$$

$$=\frac{1}{nh_{\text{bw}}}\left\{\Omega_2(k)V+o\left(1\right)\right\}$$

$$\square$$

**Theorem 6.4.** *Under assump. 3.1-3.4 and 4.1, suppose* $\|\hat{h}-h\|_2=o(1)$, $\|\hat{q}-q\|_2=o(1)$ *and* $\|\hat{h}-h\|_2\|\hat{q}-q\|_2=o((nh_{\text{bw}})^{-1/2})$, $nh_{\text{bw}}^5=O(1)$, $nh_{\text{bw}}\to\infty$, $h_0(a,w,x),p(a,z|w,x)$ *and*

$p(a, w|z, x)$ are twice continuously differentiable wrt $a$ as well as $h_0, q_0, \hat{h}, \hat{q}$ are uniformly bounded. Then for any $a$, we have the following for the bias and variance of the PKDR estimator given Eq. 7:

$$\text{Bias}(\hat{\beta}(a)) := \mathbb{E}[\hat{\beta}(a)] - \beta(a) = \frac{h_{\text{bw}}^2}{2}\kappa_2(K)B + o((nh_{\text{bw}})^{-1/2}), \text{Var}[\hat{\beta}(a)] = \frac{\Omega_2(K)}{nh_{\text{bw}}}(V + o(1)),$$

where $B = \mathbb{E}[q_0(a, Z, X)[2\frac{\partial}{\partial A}h_0(a, W, X)\frac{\partial}{\partial A}p(a, W \mid Z, X) + \frac{\partial^2}{\partial A^2}h_0(a, W, X)]]$, $V = \mathbb{E}[\mathbb{I}(A = a)q_0(a, Z, X)^2(Y - h_0(a, W, X))^2]$.

*Proof.* By Theorem. E.7 and E.8, we completed the proof. If we want to optimize the bias-variance tradeoff of the asymptotic mean squared error, we choose the optimal bandwidth $h_{\text{bw}}$ such that neither term dominates the other.

$$\text{MSE}\left(\hat{\beta}(a) - \beta(a)\right) = \text{Bias}^2 + \text{Variance}$$
$$= \frac{h_{\text{bw}}^4}{4}(\kappa(K)B)^2 + \frac{1}{nh_{\text{bw}}}\Omega(K)V + o\left(\frac{1}{nh_{\text{bw}}}\right)$$

Optimizing the leading terms of the asymptotic MSE with respect to the bandwidth $h_{\text{bw}}$:

$$\frac{\partial}{\partial h_{\text{bw}}}\text{MSE} = (\kappa(K)B)^2 h_{\text{bw}}^3 - \frac{\Omega(K)V}{nh_{\text{bw}}^2} = 0$$

Therefore, we can select the optimal bandwidth is $h_{\text{bw}} = O(n^{-1/5})$ in terms of the mean squared error (MSE) that converges at the rate of $O(n^{-4/5})$. □

### E.5 Consistency of the Estimator

**Theorem E.9.** *Under assump. 3.1-3.4 and 4.1, suppose $h_0(a, w, x)$ and $p(a, w|z, x)$ are twice continuously differentiable wrt $a$ as well as $h_0, q_0, \hat{h}, \hat{q}$ are uniformly bounded. Then, for some universal constants $c_1$ and $c_2$, with probability $1 - \eta$, the PKDR estimator given Eq. 7 error is bounded by:*

$$\left|\beta(a) - \hat{\beta}(a)\right| \leq \left\|\mathbb{I}(A = a)\left(h_0 - \hat{h}\right)\right\|_2 \|\text{proj}(\hat{q} - q_0)\|_2 + c_1\sqrt{\frac{\log(c_2/\eta)}{n}} + \frac{h_{\text{bw}}^2}{2}\kappa_2(K)R + o\left(h_{\text{bw}}^2\right)$$

*where $R = \mathbb{E}\left[\hat{q}(a, Z, X)\left[2\frac{\partial}{\partial A}h_0(a, W, X)\frac{\partial}{\partial A}p(a, W \mid Z, X) + \left(\frac{\partial^2}{\partial A^2}h_0(a, W, X)\right)\right]\right]$.*

*Proof.* From the relationship between causal effect and nuisance function, we have

$$\left|\beta(a) - \hat{\beta}(a)\right| = \left|\mathbb{E}\left[\mathbb{I}(A = a)q_0(a, Z, X)(Y - h_0(a, W, X)) + h_0(a, W, X)\right] - \hat{\beta}(a)\right|$$
$$\leq \underbrace{\left|\mathbb{E}\left[\mathbb{I}(A = a)q_0(Y - h_0) + h_0\right] - \mathbb{E}\left[\mathbb{I}(A = a)\hat{q}\left(Y - \hat{h}\right) + \hat{h}\right]\right|}_{(\mathbf{I})}$$
$$+ \underbrace{\left|\mathbb{E}\left[(\mathbb{I}(A = a) - K_{h_{\text{bw}}}(A - a))\hat{q}\left(Y - \hat{h}\right)\right]\right|}_{(\mathbf{II})}$$
$$+ \underbrace{\left|\mathbb{E}\left[K_{h_{\text{bw}}}(A - a)\hat{q}\left(Y - \hat{h}\right) + \hat{h}\right] - \hat{\beta}(a)\right|}_{(\mathbf{III})}$$

where $h_0 = h_0(a, W, X), q_0 = q_0(a, Z, X), \hat{h} = \hat{h}(a, W, X)$ and $\hat{q} = \hat{q}(a, Z, X)$.

We can bound each term as follows. For the first term, we have

$$(\mathbf{I}) = \mathbb{E}\left[\mathbb{I}(A = a)\left\{q_0(Y - h_0) - \hat{q}\left(Y - \hat{h}\right)\right\}\right] - \mathbb{E}\left[h_0 - \hat{h}\right]$$
$$= \mathbb{E}\left[\mathbb{I}(A = a)Y(q_0 - \hat{q})\right] + \mathbb{E}\left[\mathbb{I}(A = a)\left\{\hat{q}\hat{h} - q_0h_0\right\}\right] - \mathbb{E}\left[h_0 - \hat{h}\right]$$
$$\stackrel{(1)}{=} \mathbb{E}\left[\mathbb{I}(A = a)h_0(q_0 - \hat{q})\right] + \mathbb{E}\left[\mathbb{I}(A = a)\left\{\hat{q}\hat{h} - q_0h_0\right\}\right] - \mathbb{E}\left[\mathbb{I}(A = a)q_0\left(h_0 - \hat{h}\right)\right]$$
$$= \mathbb{E}\left[\mathbb{I}(A = a)(q_0 - \hat{q})\left(h_0 - \hat{h}\right)\right] \stackrel{(2)}{\leq} \left\|\mathbb{I}(A = a)\left(h_0 - \hat{h}\right)\right\|_2 \|\text{proj}(\hat{q} - q_0)\|_2$$

where (2) is derived from Cauchy's inequality and (1) is derived from

$$
\begin{aligned}
\mathbb{E}\left[\mathbb{I}\left(A=a\right)Y\left(q_0-\hat{q}\right)\right] &= \mathbb{E}\left[\mathbb{I}\left(A=a\right)\left(q_0-\hat{q}\right)\mathbb{E}\left[Y|A,Z,X\right]\right] \\
&= \mathbb{E}\left[\mathbb{I}\left(A=a\right)\left(q_0-\hat{q}\right)\mathbb{E}\left[h_0\left(A,W,X\right)|A,Z,X\right]\right] \\
&= \mathbb{E}\left[\mathbb{I}\left(A=a\right)\left(q_0-\hat{q}\right)h_0\right]
\end{aligned}
$$

$$
\begin{aligned}
\mathbb{E}\left[h_0-\hat{h}\right] &= \int\left(h_0-\hat{h}\right)\frac{p\left(a,w,x\right)}{p\left(a|w,x\right)}d\mu(w,x) \\
&= \int\left(h_0-\hat{h}\right)p\left(a,w,x\right)\mathbb{E}\left[q_0\left(a,Z,X\right)|A,W,X\right]d\mu(w,x) \\
&= \mathbb{E}\left[\mathbb{I}\left(A=a\right)\left(h_0-\hat{h}\right)q_0\right]
\end{aligned}
$$

For the second term, we have

$$
\begin{aligned}
&\mathbb{E}\left[K_{h_{\mathrm{bw}}}\left(A-a\right)\hat{q}\left(a,Z,X\right)\left(Y-\hat{h}\left(a,W,X\right)\right)\right] \\
=&\mathbb{E}\left[K_{h_{\mathrm{bw}}}\left(A-a\right)\hat{q}\left(a,Z,X\right)\mathbb{E}\left[\left(Y-\hat{h}\left(a,W,X\right)\right)|A,Z,X\right]\right] \\
=&\mathbb{E}\left[K_{h_{\mathrm{bw}}}\left(A-a\right)\hat{q}\left(a,Z,X\right)\left(h_0\left(A,W,X\right)-\hat{h}\left(a,W,X\right)\right)\right] \\
=&\mathbb{E}\left[\hat{q}\left(a,Z,X\right)\mathbb{E}\left[K_{h_{\mathrm{bw}}}\left(A-a\right)\left(h_0\left(A,W,X\right)-\hat{h}\left(a,W,X\right)\right)|Z,X\right]\right] \\
=&\mathbb{E}\left[\hat{q}\left(a,Z,X\right)\int K\left(a'-a\right)\left(h_0\left(a',w,x\right)-\hat{h}\left(a,w,x\right)\right)p\left(w,a'|z,x\right)d\mu\left(w,a'\right)\right] \\
=&\mathbb{E}\left[\hat{q}\left(a,Z,X\right)\int K\left(u\right)\underbrace{\left(h_0\left(a+h_{\mathrm{bw}}u,w,x\right)-\hat{h}\left(a,w,x\right)\right)p\left(w,a+h_{\mathrm{bw}}u|z,x\right)}_{(\star)}d\mu\left(w,u\right)\right]
\end{aligned}
$$

where the last line holds from $a' = h_{\mathrm{bw}}u + a$. Consider Taylor expansion of $h_0\left(a,w,x\right)$ and $p\left(a,w\mid z,x\right)$ around $A=a$:

$$
p\left(h_{\mathrm{bw}}u+a,w\mid z,x\right)-p\left(a,w\mid z,x\right)=h_{\mathrm{bw}}u\frac{\partial}{\partial A}p\left(a,w\mid z,x\right)+O\left(h_{\mathrm{bw}}^2\right)
$$

$$
\begin{aligned}
h_0\left(a+h_{\mathrm{bw}}u,w,x\right)-h_0\left(a,w,x\right)=&h_{\mathrm{bw}}u\left(\frac{\partial}{\partial A}h_0\left(a,w,x\right)\right) \\
&+\frac{\left(h_{\mathrm{bw}}u\right)^2}{2}\left(\frac{\partial^2}{\partial A^2}h_0\left(a,w,x\right)\right)+O\left(h_{\mathrm{bw}}^3\right)
\end{aligned}
$$

Therefore, we have

$$
\begin{aligned}
(\star)=&\left(h_0\left(a,w,x\right)-\hat{h}\left(a,w,x\right)\right)p\left(a,w\mid z,x\right) \\
&+\left(h_0\left(a,w,x\right)-\hat{h}\left(a,w,x\right)\right)h_{\mathrm{bw}}u\frac{\partial}{\partial A}p\left(a,w\mid z,x\right) \\
&+h_{\mathrm{bw}}u\left(\frac{\partial}{\partial A}h_0\left(a,w,x\right)\right)p\left(a,w\mid z,x\right) \\
&+\frac{\left(h_{\mathrm{bw}}u\right)^2}{2}\left(\frac{\partial^2}{\partial A^2}h_0\left(a,w,x\right)\right)p\left(a,w\mid z,x\right) \\
&+h_{\mathrm{bw}}u\left(\frac{\partial}{\partial A}h_0\left(a,w,x\right)\right)h_{\mathrm{bw}}u\frac{\partial}{\partial A}p\left(a,w\mid z,x\right) \\
&+\frac{\left(h_{\mathrm{bw}}u\right)^2}{2}\left(\frac{\partial^2}{\partial A^2}h_0\left(a,w,x\right)\right)h_{\mathrm{bw}}u\frac{\partial}{\partial A}p\left(a,w\mid z,x\right)+O\left(h_{\mathrm{bw}}^3\right)
\end{aligned}
$$

Then, we can compute the conditional expectation by integrating the approximation to the density term by term. Here, $\kappa_j(K)$ represents the jth kernel moment, defined as $\kappa_j(K)=\int u^j K(u)du$. It's

important to note that for a symmetric kernel, the odd-order moments integrate to 0. Therefore, we have

$$\mathbb{E}\left[K_{h_{\mathrm{bw}}}\left(A-a\right)\hat{q}\left(a,Z,X\right)\left(Y-\hat{h}\left(a,W,X\right)\right)\right]$$

$$=\mathbb{E}\left[\hat{q}\left(a,Z,X\right)\int K\left(u\right)\left(h_0\left(a+h_{\mathrm{bw}}u,w,x\right)-\hat{h}\left(a,w,x\right)\right)p\left(a+h_{\mathrm{bw}}u,w\mid z,x\right)\mathrm{d}\mu\left(u,w\right)\right]$$

$$=h_{\mathrm{bw}}^2\kappa_2(K)\mathbb{E}\left[\hat{q}\left(a,Z,X\right)\left[\frac{\partial}{\partial A}h_0\left(a,W,X\right)\frac{\partial}{\partial A}p\left(a,W\mid Z,X\right)+\frac{1}{2}\left(\frac{\partial^2}{\partial A^2}h_0\left(a,W,X\right)\right)\right]\right]$$

$$+\mathbb{E}\left[\mathbb{I}\left(A=a\right)\hat{q}\left(a,Z,X\right)\left(Y-\hat{h}\left(a,W,X\right)\right)\right]+o\left(h_{\mathrm{bw}}^2\right)$$

Therefore, we obtain

$$(\mathbf{II})=\left|\mathbb{E}\left[\left(\mathbb{I}\left(A=a\right)-K_{h_{\mathrm{bw}}}\left(A-a\right)\right)\hat{q}\left(Y-\hat{h}\right)\right]\right|$$

$$=\frac{h_{\mathrm{bw}}^2}{2}\kappa_2(K)R+o\left(h_{\mathrm{bw}}^2\right)$$

where $R=\mathbb{E}\left[\hat{q}\left(a,Z,X\right)\left[2\frac{\partial}{\partial A}h_0\left(a,W,X\right)\frac{\partial}{\partial A}p\left(a,W\mid Z,X\right)+\left(\frac{\partial^2}{\partial A^2}h_0\left(a,W,X\right)\right)\right]\right]$.

The third terms are upper-bounded by Bernstein inequality. This concludes

$$\left|\beta(a)-\hat{\beta}(a)\right|\le\left\|\mathbb{I}\left(A=a\right)\left(h_0-\hat{h}\right)\right\|_2\left\|\mathrm{proj}\left(\hat{q}-q_0\right)\right\|_2+c_1\sqrt{\frac{\log\left(c_2/\eta\right)}{n}}+\frac{h_{\mathrm{bw}}^2}{2}\kappa_2(K)R+o\left(h_{\mathrm{bw}}^2\right)$$

$$\square$$

*Remark* E.10. According to Thm. 6.2 and E.6, we have $\|\hat{h}-h_0\|_2=O(n^{-1/4})$ and $\|\hat{q}-q_0\|_2=O(n^{-1/4})$. Therefore the order of the estimation error is controlled by $h_{\mathrm{bw}}$. From Thm. 6.4, we know that the optimal bandwidth is $h_{\mathrm{bw}}=O(n^{-1/5})$ in terms of estimator error that converges at the rate of $O(n^{-2/5})$. Note that this rate is slower than the optimal rate $O(n^{-1/2})$, which is a reasonable sacrifice to handle continuous treatment within the proximal causal framework and agrees with existing studies (Kennedy et al., 2017; Colangelo & Lee, 2020).

## F    COMPUTATION DETAILS

We can consider the nuisance/bridge function class $\mathcal{Q}$ or $\mathcal{H}$ and the dual/critic functional class $\mathcal{M}$ or $\mathcal{G}$ are the RKHS class. The inner maximization in Eq. 8 and 9 may no longer have closed-form solutions with the RKHS norm constraints. Similar to Dikkala et al. (2020), we consider the following optimization problem

$$\min_{q \in \mathcal{Q}} \max_{m \in \mathcal{M}} \frac{1}{n} \sum_i \left( q(a_i, z_i, x_i) - \frac{1}{p(a_i|w_i, x_i)} \right) m(a_i, w_i, x_i) - \lambda_m \|m\|_{2,n}^2 - \gamma_m \|m\|_{\mathcal{M}}^2 + \gamma_q \|q\|_{\mathcal{Q}}^2$$

$$\min_{h \in \mathcal{H}} \max_{g \in \mathcal{G}} \frac{1}{n} \sum_i (y_i - h(w_i, a_i, x_i)) g(a_i, z_i, x_i) - \lambda_g \|g\|_{2,n}^2 - \gamma_g \|g\|_{\mathcal{M}}^2 + \gamma_h \|h\|_{\mathcal{Q}}^2$$

**Proposition F.1.** *Suppose $\mathcal{M}$ and $\mathcal{G}$ are RKHS spaces with kernel $K_{\mathcal{M}}$ and $K_{\mathcal{G}}$ equipped with the canonical RKHS norm, then for any $q, h$, we have*

$$\max_{m \in \mathcal{M}} \Phi_q^n - \lambda_m \|m\|_{2,n}^2 - \gamma_m \|m\|_{\mathcal{M}}^2 = \frac{1}{4\gamma_m} \psi_{q,n}^\top K_{\mathcal{M},n} \left( \frac{\lambda_m}{\gamma_m} \frac{1}{n} K_{\mathcal{M},n} + \gamma_m I \right)^{-1} \psi_{q,n}$$

$$= \frac{1}{4\gamma_m} \psi_n^\top K_{\mathcal{M},n}^{1/2} \left( \frac{\lambda_m}{\gamma_m} \frac{1}{n} K_{\mathcal{M},n} + \gamma_m I \right)^{-1} K_{\mathcal{M},n}^{1/2} \psi_n$$

$$\max_{g \in \mathcal{G}} \Phi_h^n - \lambda_g \|g\|_{2,n}^2 - \gamma_g \|g\|_{\mathcal{G}}^2 = \frac{1}{4\gamma_g} \psi_{h,n}^\top K_{\mathcal{G},n} \left( \frac{\lambda_g}{\gamma_g} \frac{1}{n} K_{\mathcal{G},n} + \gamma_g I \right)^{-1} \psi_{h,n}$$

$$= \frac{1}{4\gamma_g} \psi_{h,n}^\top K_{\mathcal{G},n}^{1/2} \left( \frac{\lambda_g}{\gamma_g} \frac{1}{n} K_{\mathcal{G},n} + \gamma_g I \right)^{-1} K_{\mathcal{G},n}^{1/2} \psi_{h,n}^\top$$

*where $K_{\mathcal{M},n} = (K_{\mathcal{M}}(a_i, w_i, x_i, a_j, w_j, x_j))_{i,j=1}^n$, $K_{\mathcal{G},n} = (K_{\mathcal{G}}(a_i, z_i, x_i, a_j, z_j, x_j))_{i,j=1}^n$ the empirical kernel matrix and $\psi_{q,n} = (\frac{1}{n}(q(a_i, z_i, x_i) - \frac{1}{p(a_i|w_i, x_i)}))_{i=1}^n$, $\psi_{h,n} = \left( \frac{1}{n}(y_i - h(a_i, w_i, x_i)) \right)_{i=1}^n$.*

*Proof.* By the generalized representer theorem of Schölkopf et al. (2001), implies that an optimal solution of the constrained problem takes the form

$$m(a, w, x) = \sum_{i=1}^n \alpha_i K_{\mathcal{M}}(a_i, w_i, x_i, a, w, x).$$

We denote $K_{\mathcal{M},n} = (K_{\mathcal{M}}(a_i, w_i, x_i, a_j, w_j, x_j))_{i,j=1}^n$ the empirical kernel matrix. And we have $\|m\|_{\mathcal{M}}^2 = \alpha^\top K_{\mathcal{M},n} \alpha$, $f(z_i) = e_i^\top K_{\mathcal{M},n} \alpha$ and $\|m\|_{2,n}^2 = \frac{1}{n} \alpha^\top K_{\mathcal{M},n}^2 \alpha$. Thus the penalized problem is equivalent to the finite dimensional maximization problem:

$$\max_{\alpha \in \mathbb{R}^n} \psi_{q,n}^\top K_{\mathcal{M},n} \alpha - \alpha^\top \left( \frac{\lambda_m}{n} K_{\mathcal{M},n} + \gamma_m I \right) K_{\mathcal{M},n} \alpha,$$

where $\psi_{q,n} = \left( \frac{1}{n} \left( q(a_i, z_i, x_i) - \frac{1}{p(a_i|w_i, x_i)} \right) \right)_{i=1}^n$. By taking the first order condition, the latter has a closed form optimizer of:

$$\alpha = \frac{1}{2\gamma_m} \left( \frac{\lambda_m}{\gamma_m} \frac{1}{n} K_{\mathcal{M},n} + I \right)^{-1} \psi_{q,n}$$

and optimal value of:

$$\frac{1}{4\gamma_m} \psi_n^\top K_{\mathcal{M},n} \left( \frac{\lambda_m}{\gamma_m} \frac{1}{n} K_{\mathcal{M},n} + I \right)^{-1} \psi_{q,n} = \frac{1}{4\gamma_m} \psi_n^\top K_{\mathcal{M},n}^{1/2} \left( \frac{\lambda_m}{\gamma_m} \frac{1}{n} K_{\mathcal{M},n} + I \right)^{-1} K_{\mathcal{M},n}^{1/2} \psi_{q,n}$$

Similarly, we have

$$\max_{g \in g} \Phi_h^n - \lambda_g \|g\|_{2,n}^2 - \gamma_g \|g\|_{\mathcal{G}}^2 = \frac{1}{4\gamma_g} \psi_{h,n}^\top K_{\mathcal{G},n} \left( \frac{\lambda_g}{\gamma_g} \frac{1}{n} K_{\mathcal{G},n} + I \right)^{-1} \psi_{h,n}$$

$$= \frac{1}{4\gamma_g} \psi_{h,n}^\top K_{\mathcal{G},n}^{1/2} \left( \frac{\lambda_g}{\gamma_g} \frac{1}{n} K_{\mathcal{G},n} + I \right)^{-1} K_{\mathcal{G},n}^{1/2} \psi_{h,n}^\top$$

where $\psi_{h,n} = \left( \frac{1}{n}(y_i - h(a_i, w_i, x_i)) \right)_{i=1}^n$. $\qquad \square$

If further $\mathcal{Q}$ and $\mathcal{H}$ are RKHS, we can obtain the closed form solution about the outer maximization, by solving

$$\hat{q} = \arg\min_{q \in \mathcal{Q}} \psi_{q,n}^\top K_{\mathcal{M},n} \left( \frac{\lambda_m}{\gamma_m} \frac{1}{n} K_{\mathcal{M},n} + I \right)^{-1} \psi_{q,n} + 4\gamma_m \gamma_q \|q\|_{\mathcal{Q}}^2$$

$$\hat{h} = \arg\min_{h \in \mathcal{H}} \psi_{h,n}^\top K_{\mathcal{G},n} \left( \frac{\lambda_g}{\gamma_g} K_{\mathcal{G},n} + I \right)^{-1} \psi_{h,n} + 4\gamma_g \gamma_h \|h\|_{\mathcal{H}}^2$$

Again by the representation theorem, we have

$$\hat{h}(\cdot) = \sum_i \hat{\alpha}_i k\left((a_i, w_i, x_i), \cdot\right), \quad \hat{q}(\cdot) = \sum_i \hat{\beta}_i k\left((a_i, z_i, x_i), \cdot\right)$$

where

$$\hat{\alpha} = \left(K_{\mathcal{H},n} G_h K_{\mathcal{H},n} + 4\gamma_h \gamma_g K_{\mathcal{H},n}\right)^{-1} K_{\mathcal{H},n} G_h y_i$$

$$\hat{\beta} = \left(K_{\mathcal{Q},n} M_q K_{\mathcal{Q},n} + 4\gamma_q \gamma_m K_{\mathcal{Q},n}\right)^{-1} K_{\mathcal{Q},n} M_q \frac{1}{p(a_i|w_i, x_i)}$$

for $G_h = K_{\mathcal{G},n}^{1/2} \left( \frac{\lambda_g}{\gamma_g} \frac{1}{n} K_{\mathcal{G},n} + I \right)^{-1} K_{\mathcal{G},n}^{1/2}$ and $M_q = K_{\mathcal{M},n}^{1/2} \left( \frac{\lambda_m}{\gamma_m} \frac{1}{n} K_{\mathcal{M},n} + I \right)^{-1} K_{\mathcal{M},n}^{1/2}$.

There are several tuning parameters in the estimation of $h_0$ and $q_0$. We accept the tricks and recommendation defaults by Dikkala et al. (2020). The following parameters will be used to determine.

$$\frac{\lambda_g}{\gamma_g}(n) = \frac{5}{n^{0.4}} \tag{29}$$

$$\gamma_h \gamma_g(s, n) = \frac{s}{2} \left( \frac{\lambda_g}{\gamma_g}(n) \right)^4 \tag{30}$$

For $\frac{\lambda_m}{\gamma_m}$ and $\gamma_q \gamma_m$, we also choose parameters like this.

# G  ADDITIONAL EXPERIMENTS

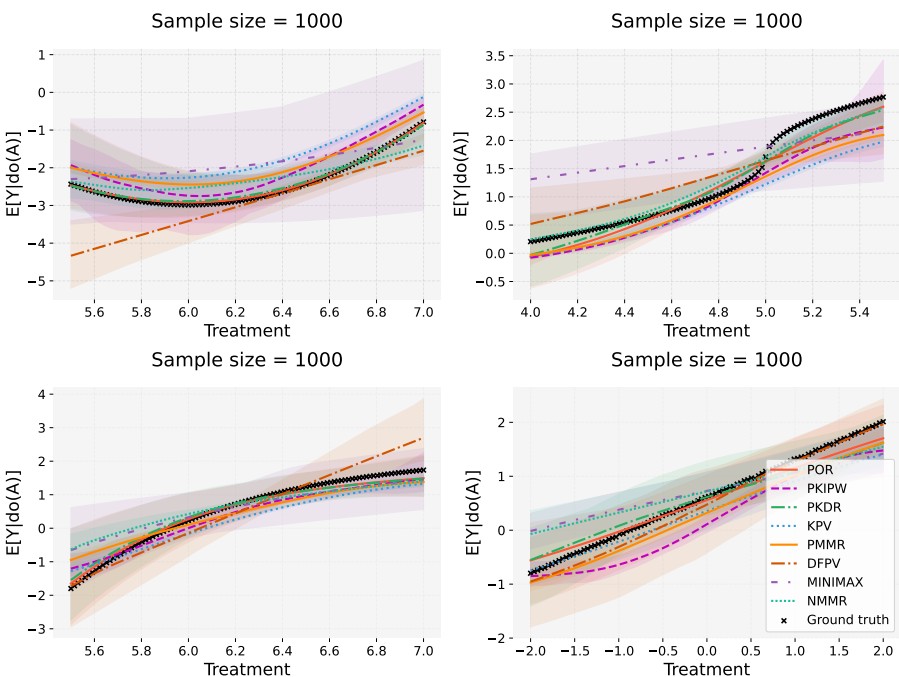

Figure 4: ATE comparison of different methods across various methods on three different data-generating mechanisms; Left Top: ATE comparison using 1000 samples in the Scenario 1; Right Top: ATE comparison using 1000 samples in the Scenario 2; Left Bottom: ATE comparison using 1000 samples in the Scenario 3; Right Bottom: ATE comparison using 1000 samples in the Times series data.

In this section, we consider three more synthetic settings introduced in Hu et al. (2023), as well as the times-series setting introduced in Miao et al. (2018b) that satisfies the proximal causality framework. Similar to Tab. 1, our methods are comparable or better than others.

**Implementation Details.**  In the PKIPW and PKDR estimators, we choose the second-order Epanechnikov kernel, with bandwidth $h_{\mathrm{bw}} = c\hat{\sigma}_A n^{-1/5}$ with estimated std $\hat{\sigma}_A$ and the hyperparameter $c = 1.5$. For policy estimation, we employ the KDE in the two datasets. The rest of the implementation details are consistent with the experiments in the text.

**Evaluation metrics.** We report the causal Mean Squared Error (cMSE) across 100 equally spaced points in the range of $\mathrm{supp}(A)$: $\mathrm{cMSE} := \frac{1}{100}\sum_{i=1}^{100}(\mathbb{E}[Y^{a_i}] - \hat{\mathbb{E}}[Y^{a_i}])^2$. Here, we respectively take $\mathrm{supp}(A) := [5.5, 7], [4, 5.5], [5.5, 7], [-2, 2]$ in three synthetic data and the times series data. The truth $\mathbb{E}[Y^a]$ is derived through Monte Carlo simulations comprising 10,000 replicates of data generation for each $a$.

## G.1  EXPERIMENTS WITH DIFFERENT DATA GENERATING PROCESS

**Data generation.** We consider three different data-generating mechanisms (Hu et al., 2023). Under each of scenarios, we simulate the continuous unmeasured confounder $U$ following a normal distribution with mean 1 and variance 0.2, denoted by $U \sim N(1, 0.2)$. Similarly, we simulate two type of proxy variables $W|U \sim N(1 - 2 \cdot U, 0.2)$ and $Z|U \sim N(-1 + 1.5 \cdot U, 0.2)$. The three scenarios vary according to the data generation process based on the models for the outcome $Y|A, U$, and for the treatment $A|U$.

- **Scenario 1.** We assume that the true distribution of the outcome $Y$ is a parabola, i.e., the outcome is a second-order regression function of the treatment:
$$Y|A, U \sim N(-10 + 2.2 \cdot (A - 6)^2 + 4 \cdot U_i, 0.2)$$

and $A|U \sim N(2.5 + 4 \cdot U, 0.2)$.

- **Scenario 2.** We assume that the true distribution of the outcome $Y$ has a sigmoidal shape:

$$Y|A, U \sim N(1.5 + \text{sign}(A - 5) \cdot \sqrt{|A - 5|} + 1.7 \cdot U, 0.05),$$

where sign is the sign function such that it is equal to 1 when $a \geq 0$ and -1 otherwise. We assume $A|U \sim N(1 + 4 \cdot U, 0.2)$.

- **Scenario 3.** We assume that the true distribution of the outcome $Y$ is monotonically increasing with a non-linear relationship with both variables $A$ and $U$:

$$Y|A, U \sim N(-2 \cdot e^{-1.4 \cdot (A-6)} + 0.8 \cdot e^U, 0.2).$$

We assume $A \mid U \sim N(2.5 + 4 \cdot U, 0.2)$

**Results.** We report the mean and the standard deviation (std) of cMSE over 20 times across four scenarios, as depicted in Fig. 4 and Tab. 3. For each scenario, we take $n = 1,000$. We can see that the PKIPW method suffers from large errors in scenarios 1 and 2 while performing well in scenario 3, where the treatment-inducing proxy $Z$ is misspecified. However, the PKDR method still performs well due to its doubly robust. In addition, we find that the MINIMAX method does not perform well because it requires more samples to fit the neural network.

Table 3: cMSE of all methods on three different data-generating mechanisms.

| Dataset | | Size | PMMR | KPV | DFPV | MINIMAX | NMMR | POR | **PKIPW** | **PKDR** |
|---|---|---|---|---|---|---|---|---|---|---|
| Hu et al. (2023) | Scenario 1 | 1000 | $0.25_{\pm 0.05}$ | $0.25_{\pm 0.05}$ | $0.59_{\pm 0.35}$ | $1.45_{\pm 1.32}$ | $0.17_{\pm 0.09}$ | $0.16_{\pm 0.23}$ | $0.29_{\pm 0.12}$ | $\mathbf{0.16_{\pm 0.22}}$ |
| | Scenario 2 | 1000 | $0.16_{\pm 0.02}$ | $0.16_{\pm 0.02}$ | $0.22_{\pm 0.17}$ | $0.88_{\pm 0.29}$ | $\mathbf{0.05_{\pm 0.04}}$ | $0.08_{\pm 0.07}$ | $0.15_{\pm 0.06}$ | $0.07_{\pm 0.06}$ |
| | Scenario 3 | 1000 | $0.10_{\pm 0.02}$ | $0.10_{\pm 0.02}$ | $0.28_{\pm 0.39}$ | $0.45_{\pm 0.29}$ | $0.21_{\pm 0.10}$ | $0.22_{\pm 0.20}$ | $\mathbf{0.09_{\pm 0.03}}$ | $0.21_{\pm 0.19}$ |
| Time Series | | 500 | $0.11_{\pm 0.04}$ | $0.13_{\pm 0.12}$ | $0.20_{\pm 0.14}$ | $0.21_{\pm 0.09}$ | $0.18_{\pm 0.12}$ | $0.10_{\pm 0.14}$ | $0.21_{\pm 0.05}$ | $\mathbf{0.09_{\pm 0.12}}$ |
| | | 1000 | $0.10_{\pm 0.05}$ | $\mathbf{0.08_{\pm 0.07}}$ | $0.16_{\pm 0.21}$ | $0.22_{\pm 0.06}$ | $0.18_{\pm 0.10}$ | $0.12_{\pm 0.12}$ | $0.20_{\pm 0.06}$ | $0.12_{\pm 0.10}$ |

## G.2 EXPERIMENTS FOR TIME SERIES DATA

**Data generation.** We follow Miao et al. (2018b) to generate data.

$$U_i = \xi U_{i-1} + (1 - \xi^2)^{1/2}\varepsilon_{1i}, \quad V_i = 0.6U_i + \varepsilon_{2i}, \quad A_i = 0.4 + 1.5V_i + \eta U_i + \varepsilon_{3i},$$
$$Y_i = 0.5 + 0.7A_i + 1.5V_i + 0.9U_i + \varepsilon_{4i}, \quad \varepsilon_{1i}, \varepsilon_{2i}, \varepsilon_{3i}, \varepsilon_{4i} \sim N(0, 1),$$

where $U_i$ is a stationary autoregressive process with autocorrelation coefficient $\xi$, and $\eta$ controls the magnitude of confounding. Here, we let $\xi = 0.8$, $\eta = 0.5$. For our proximal causal approach, we use $W_i = Y_{i-1}$ and $Z_i = A_{i+1}$ as two types of proxy variables and do not need auxiliary data.

**Results.** We report the mean and the standard deviation (std) of cMSE over 20 times across four scenarios, as depicted in Fig. 4 and Tab. 3. For each scenario, we consider two sample sizes, $n = 500$ and $n = 1,000$. As shown in Fig. 4, our PKIPW and PKDR accurately estimate the causal effect across all treatment values, making its overall cMSE comparable or better than other baselines. This result suggests the effectiveness of our methods for different scenarios.

## G.3 RATE

Due to the error introduced in kernel approximation, this is a reasonable sacrifice to handle continuous treatment within the proximal causal framework. Besides, according to Ichimura & Newey (2022), since the estimand is non-regular, therefore it may not enjoy the properties of $\sqrt{n}$-consistent and asymptotically normality. Such flexible kernel function approximation will make a non-negligible contribution to the limiting behavior of the estimator, preventing asymptotic normality and root-n consistency.

We conducted empirical numerical verification using Scenario 1 from the initial synthesis experiment outlined in Appendix G. As per the synthesis mechanism, we can easily obtain the density function

$$f(A|W) = \frac{1}{\sqrt{0.4\pi}}e^{-\frac{1}{0.4}(A-2.5+4W)^2}.$$

We compute the empirical estimator error with sample sizes $\{200, 400, 600, 800, 1,000\}$ and compare the estimator error in Figure 5. As we speculated before, the convergence rate is difficult to reach $n^{-1/2}$.

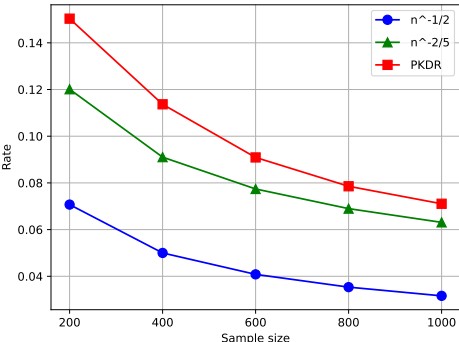

Figure 5: Empirical estimator error of the Scenario 1 in Appendix G.

# H  EXPERIMENTS

In this section, we present the data generation process of experiments and the detailed settings of hyper-parameters.

## H.1  DATA GENERATING PROCESS IN THE LOW-DIMENSIONAL SYNTHETIC EXPERIMENT

We describe the data generating mechanism in the Synthetic-Data Experiment. The generative process from Mastouri et al. (2021). Since the original data generates $Y \in [0, 1]$ and the overall trend is flat, we modify the structural equation of $Y$ to make it easier to distinguish.

$$U := [U_1, U_2], \quad U_2 \sim \text{Uniform}\,[-1, 2]$$
$$U_1 \sim \text{Uniform}\,[0, 1] - \mathbb{I}\,[0 \leq U_2 \leq 1]$$
$$W := [W_1, W_2] = [U_1 + \text{Uniform}\,[-1, 1]\,, U_2 + \varepsilon_1]$$
$$Z := [Z_1, Z_2] = [U_1 + \varepsilon_2, U_2 + \text{Uniform}[-1, 1]]$$
$$A := U_2 + \varepsilon_3$$
$$Y := 3\cos\left(2\left(0.3U_1 + 0.3U_2 + 0.2\right) + 1.5A\right) + \varepsilon_4$$

where $\{\epsilon_i\}_{i=1}^4 \sim N(0, 1)$.

## H.2  DATA GENERATING PROCESS IN THE HIGH-DIMENSIONAL SYNTHETIC EXPERIMENT

For $X \in \mathbb{R}^{\dim(X)}$, $Z \in \mathbb{R}^{\dim(Z)}$, $W \in \mathbb{R}^{\dim(W)}$ and $(A, D) \in \mathbb{R}$, we first generate the unobserved noise:

$$\{\epsilon_i\}_{i \in [3]} \overset{i.i.d}{\sim} N(0, 1), \quad \nu_z \sim \text{Uniform}[-1, 1]^{\dim(Z)}, \quad \nu_w \sim \text{Uniform}[-1, 1]^{\dim(W)}$$

Next, we generate the following data structure

- For unobserved confounders $U$, we have

$$U_z = \epsilon_1 + \epsilon_3, \quad U_w = \epsilon_2 + \epsilon_3$$

- For two types of proxies $Z$ and $W$, we have

$$Z = \nu_z + 0.25 \cdot U_z \cdot \mathbf{1}_{dim(Z)}, \quad W = \nu_w + 0.25 \cdot U_w \cdot \mathbf{1}_{dim(W)}$$

  where $\mathbf{1}_p \in \mathbb{R}^p$ is the vector of ones of length $p$.
- For covariates $X$, we have

$$X \sim N(0, \Sigma), \text{ where } \Sigma \in \mathbb{R}^{\dim(X) \times \dim(X)}, \Sigma_{ii} = 1 \text{ and } \Sigma_{ij} = \frac{1}{2} \cdot \mathbb{I}\{|i - j| = 1\} \text{ for } i \neq j.$$

- For treatment $A$, we have

$$A = \Lambda(3X^\top \beta_x + 3Z^\top \beta_z) + 0.25 \cdot U_w,$$

  where $\beta_x \in \mathbb{R}^{\dim(X)}$ and $\beta_z \in \mathbb{R}^{\dim(Z)}$ are quadratically decaying coefficients, e.g. $[\beta_x]_j = j^{-2}$. $\Lambda$ is the truncated logistic link function $\Lambda(t) = (0.9 - 0.1)\frac{\exp(t)}{1+\exp(t)} + 0.1$.
- For outcome $Y$, we have

$$Y = \theta_0^{\text{ATE}}(A) + 1.2(X^\top \beta_x + W^\top \beta_w) + AX_1 + 0.25 \cdot U_z,$$

  where $\beta_w \in \mathbb{R}^{\dim(W)}$ are quadratically decaying coefficients, e.g. $[\beta_w]_j = j^{-2}$.

Follow Colangelo & Lee (2020), we use the quadratic design, $\theta_0^{\text{ATE}}(a) = a^2 + 1.2a$.

## H.3  LEGALIZED ABORTION AND CRIME

In the Abortion and Criminality dataset, as described in the reference Woody et al. (2020), the key variables are as follows:

- Treatment Variable $A$: Effective abortion rate;

- Outcome variable $Y$: Murder rate;
- Treatment-inducing proxy $Z$: Generosity towards families with dependent children;
- Outcome-inducing proxy $W$: Beer consumption per capita, log-prisoner population per capita, and concealed weapons laws.

We take the remaining variables as the unobserved confounding variables $U$. Following Mastouri et al. (2021), the ground-truth value of $\beta(a)$ is taken from the generative model fitted to the data.

The dataset is available at `https://github.com/yuchen-zhu/kernel_proxies/tree/main/data/sim_1d_no_x`.

### H.4 HYPERPARAMETERS SELECTION

In all our numerical studies, RKHSs $\mathcal{G}, \mathcal{H}, \mathcal{M}, \mathcal{Q}$ are equipped with Gaussian kernels

$$K(x_1, x_2) = \exp\{\gamma \|x_1 - x_2\|_2^2\}.$$

The median heuristic bandwidth parameter $\gamma^{-1} = \text{median}\{\|x_i - x_j\|_2^2\}_{i<j \in I}$ for indices subset $I \subset \{1, \dots, n\}$. For the regularization coefficient, we automatically select it according to Eq. 29 and 30.

For KDE, we also choose the Gaussian kernel. For bandwidth, we employ three fold cross-validation, where the bandwidth is chosen as 20 values uniformly distributed in logarithmic space between $10^{-0.1}$ and 10 raised to $10^0$.

For CNFs, we recommend using the package $\mathrm{probaforms}$, where the prior distributions is multivariate normal distribution.

Table 4: CNFs-block

| Layer | Configuration |
|---|---|
| 1 | Input$(A_i, W_i, X_i)$ |
| 2 | FC(in-dim, 128), ReLU |
| 3 | FC(128, 64), ReLU |
| 3 | FC(64, 32) |

We stack four CNFs-block and finally solve the density function.

Table 5: Hyperparameters for CNFs.

| Hyperparameter | |
|---|---|
| Learning rate | 1e-4 |
| Epochs | 500 |
| Batch_size | 512 |
| Weight_decay | 1e-4 |

For the KPV method, we used the Gaussian kernel where the bandwidth is determined by the median trick. We select the regularizers $\lambda_1 = \lambda_2 = 0.005$.

For the PMMR method, we used the Gaussian kernel where the bandwidth is determined by the median trick. We select the regularizers $\lambda_1 = \lambda_2 = 0.1$.

For the DFPV, we optimize the model using Adam with learning rate = 0.001, $\beta_1 = 0.9$, $\beta_2 = 0.999$ and $\varepsilon = 10^{-8}$. Regularizers $\lambda_1 = \lambda_2$ are both set to 0.1 as a result of the tuning procedure.

For the MINIMAX, we used learner and adversary networks where learner l2= 1e-4,learner lr=1e-4, adversary l2 = 5e-3 and adversary lr = 5e-4.

For the NMMR, we optimize the model using Adam with learning rate =3e-3, decay=3e-6 and epoch = 10000.

