# OpenReview forum: "Doubly Robust Proximal Causal Learning for Continuous Treatments"
_ICLR.cc/2024/Conference — ICLR 2024 poster_

### Official Review · Reviewer_eqxp · 2023-10-30

**Soundness:** 3 good
**Presentation:** 3 good
**Contribution:** 3 good
**Rating:** 6
**Confidence:** 3

**Summary:**

The paper presents a kernel-based doubly robust estimator designed for proximal causal learning, particularly adept at handling continuous treatments. The authors demonstrate that this estimator is a consistent approximation of the influence function. They also introduce an innovative method for efficiently resolving nuisance functions. Additionally, the paper includes an in-depth convergence analysis in relation to the mean square error.

The paper proposes a novel and intriguing problem within the proximal causal learning framework.  Despite the motivation not being particularly compelling, the problem itself remains highly intriguing and of considerable importance.

**Strengths:**

* The paper is well-structured and flows naturally.
* This paper proposes an intriguing research topic within proximal causal inference work.
* Theoretical guarantee is also provided.

**Weaknesses:**

* The inference issue seems to be ignored without the analysis of asymptotical distribution for causal effect.

* The paper’s rationale for the calculation of the influence function appears to be lacking, as it directly selects a specific submodel. It might be more appropriate to refer to the function derived in this paper as the efficient influence function, given that the previous doubly robust estimator for binary treatment is efficient.

* The empirical coverage probability is not given. MSE only contains both bias and variance terms, which may not display the true statistic estimation acuraccy and precision.

**Questions:**

The authors have well clarified their setting and their problem is also novel. I only have several comments below.

* It will be more appropriate to call Assumption 3.2 the "latent ignorability" instead of "conditional randomization".
* Although one single optimization reduces computational burden, but asymptotical normality of causal effect estimate may not hold using the proposed kernel estimation method. So please make the motivation clear about why this paper does not focus on the inference issue.
* Why does the projected residual mean squared error make sense? Does it mean \hat{q} is consistent for q0?
* this paper seems to consider the high-dimensional setting in experiments, what about theoretical properties?

---

> ### Author Response · Authors · 2023-11-19
> **Response to Reviewer eqxp**
>
> We appreciate your efforts in reviewing our paper and constructive comments. We have revised our manuscript based on your advice. We address your concerns in the following.
>
> **Q1.** The paper’s rationale for the calculation of the influence function appears to be lacking, as it directly selects a specific submodel. It might be more appropriate to refer to the function derived in this paper as the efficient influence function, given that the previous doubly robust estimator for binary treatment is efficient.
>
> **A.** While the DR estimator for binary treatment is shown to be the effective influence function, it is unclear whether it still holds for continuous treatments without explicit computation. To verify, we calculate the influence function.
>
> **Q2.** Why does the projected residual mean squared error make sense? Does it mean $\hat{q}$ is consistent for $q_0$?
>
> **A.** The projected residual mean squared error measures how much $\hat{q}$ violate Eq. (3). This performance metric holds significant prominence in the theoretical analysis of minimax problems, as noted in references [1, 2, 3]. When the measures of the ill-posedness of inverse problems are bounded, $\hat{q}$ is consistent for $q_0$. Please refer to Remark 6.3 for details.
>
> **Q3.** Please make the motivation clear about why this paper does not focus on the inference issue about the analysis of asymptotically distribution for a causal effect.
>
> **A.** Currently, we cannot obtain the asymptotic normality due to the error introduced in kernel approximation. With this error, we can show in Theorem E.9 that our estimator is $n^{2/5}$-consistent, while the asymptotic normality means $\sqrt{n}$-consistent. Besides, according to [4], since the estimand is non-regular, therefore it may not enjoy the properties of $\sqrt{n}$-consistent and asymptotically normality. We illustrate this point through an empirical study in Appendix E.5.
>
> **Q4.** The empirical coverage probability is not given. MSE only contains both bias and variance terms, which may not display the true statistic estimation accuracy and precision.
>
> **A.** Thank you for your suggestions. We provide in Theorem E.9 that our estimator is $n^{2/5}$-consistent, which means with high probability, the error is $O(n^{-2/5})$.
>
> **Q5.** This paper seems to consider the high-dimensional setting in experiments, what about theoretical properties?
>
> **A.** In this context, we follow the setting in [5,6], in which the term "high-dimensional" merely indicates a scenario with relatively more covariates compared to those in section 7.1.1, without implying that the number of samples is smaller than the number of features. Therefore, our theory still applies to this case.
>
> [1] Dikkala, Nishanth, et al. "Minimax estimation of conditional moment models." Advances in Neural Information Processing Systems 33 (2020): 12248-12262.
>
> [2] Ghassami, AmirEmad, et al. "Minimax kernel machine learning for a class of doubly robust functionals with application to proximal causal inference." International Conference on Artificial Intelligence and Statistics. PMLR, 2022.
>
> [3] Qi, Zhengling, Rui Miao, and Xiaoke Zhang. "Proximal learning for individualized treatment regimes under unmeasured confounding." Journal of the American Statistical Association (2023): 1-14.
>
> [4] Colangelo, Kyle, and Ying-Ying Lee. "Double debiased machine learning nonparametric inference with continuous treatments." arXiv preprint arXiv:2004.03036 (2020).
>
> [5] Xu, Liyuan, Heishiro Kanagawa, and Arthur Gretton. "Deep proxy causal learning and its application to confounded bandit policy evaluation." Advances in Neural Information Processing Systems 34 (2021): 26264-26275.
>
> [6] Kompa, Benjamin, et al. "Deep learning methods for proximal inference via maximum moment restriction." Advances in Neural Information Processing Systems 35 (2022): 11189-11201.

---

### Official Review · Reviewer_sZEk · 2023-10-30

**Soundness:** 4 excellent
**Presentation:** 3 good
**Contribution:** 3 good
**Rating:** 6
**Confidence:** 4

**Summary:**

This paper introduces a method for proximal causal inference when dealing with continuous treatments. The proposed method is doubly robust in terms of estimating nuisances.

**Strengths:**

1. This paper is technically sound and strong.
2. The experimental studies are well done. A sufficient amount of empirical evidence for the proposed method is provided.

**Weaknesses:**

1. I believe that the statement of Theorem 4.5 is incorrect. More precisely, the influence function of $B(a)$ does not exist, meaning that $B(a)$ is not pathwise differentiable. However, $B(a; P^{\epsilon,h_{bw}})$ is pathwise differentiable, indicating that the influence function does exist. Therefore, to accurately state this, the term "lim" should be removed and the statement should be made with "for any $h_{bw} > 0$".
2. A practical guide is needed to solve the optimization problem in Equations (8,9) and apply these methods in practice.

**Questions:**

1. I believe that the statement of Theorem 4.5 is incorrect. More precisely, the influence function of $B(a)$ does not exist, meaning that $B(a)$ is not pathwise differentiable. However, $B(a; P^{\epsilon,h_{bw}})$ is pathwise differentiable, indicating that the influence function does exist. Therefore, to accurately state this, the term "lim" should be removed and the statement should be made with "for any $h_{bw} > 0$".
2. I understand that $q_0$ represents the maximum value for the equation in Lemma 5.1. However, I am unsure about the connection between Lemma 5.1 and Equation (8). In other words, why do we need to minimize the empirical quantity for the equation in Lemma 5.1?

---

> ### Author Response · Authors · 2023-11-19
> **Response to Reviewer sZEk**
>
> We appreciate reviewer sZEk for the time and effort in reviewing our paper. We address your concerns below.
>
> **Q1.** I believe that the statement of Theorem 4.5 is incorrect. More precisely, the influence function of $B(a)$ does not exist, meaning that $B(a)$ is not pathwise differentiable. However, $B(a,P^{\varepsilon,h_{bw}})$ is pathwise differentiable, indicating that the influence function does exist. Therefore, to accurately state this, the term `lim' should be removed and the statement should be made with for any $h_{bw}>0$.
>
> **A.** While $\beta(a,P^{\varepsilon,h_{bw}})$ is pathwise differentiable, its influence function is not of our interest. Our goal is to obtain the influence function for $\beta(a)$. To obtain this, we take the gradient with respect to $\varepsilon$ and let $h$ go to 0. Such a treatment has been similarly adopted in [1].
>
> **Q2.** A practical guide is needed to solve the optimization problem in Equations (8,9) and apply these methods in practice.
>
> **A.** Equations (8,9) are a minimax optimization problem, and we can solve it in different ways, depending on what function space class we use. For example, We can parameterize $q$ (resp. $h$) and $m$ (resp. $g$) as reproducing kernel Hilbert space (RKHS) with kernel function or neural networks. For the former, we derive their closed solutions in Appx. F. In the latter case, we can employ Generative Adversarial Networks [2]. We have appended this discussion to make it clearer.
>
> **Q3.** I understand that $q_0$ represents the maximum value for the equation in Lemma 5.1. However, I am unsure about the connection between Lemma 5.1 and Equation (8). In other words, why do we need to minimize the empirical quantity for the equation in Lemma 5.1?
>
> **A.** We have modified the description of Lemma 5.1 and added a paragraph to discuss its connection to Eq. (3) and Eq. (8). Simply speaking,  We first show that solving $q_0$ from data is equivalent to minimizing $\mathcal{L}_q(q;p):=\mathbb{E} [ \left( \mathcal{R} _q\left( q,p \right) \right) ^2 ]$ over $q$, where $\mathcal{L}_q(q;p)$ is equivalent to the maximization form in Lemma 5.1.
>
> [1] Hidehiko Ichimura and Whitney K Newey. The influence function of semiparametric estimators.
> Quantitative Economics, 13(1):29–61, 2022.
>
> [2] Goodfellow, Ian, et al. "Generative adversarial nets." Advances in neural information processing systems 27 (2014).

---

> > ### Comment · Reviewer_sZEk · 2023-12-01
> > **Response for reviews**
> >
> > I am satisfied with the answers to my questions Q2 and Q3.
> >
> > However, "Our goal is to obtain the influence function for $\beta(a)$" doesn't make sense since the influence function of $\beta(a)$ doesn't exist. I want to emphasize that it's incorrect to state "an influence function for $\beta(a)$" because it doesn't exist. Therefore, I strongly believe that Theorem 4.5 is wrong. To make it correct, $\lim_{h_{bw} \rightarrow 0}$ should be removed. In the presence of continuous treatments, the best thing we can do is to approximate $\beta(a)$ as $\beta(a,P^{\epsilon,h_{bw}}$, then working on the influence function of $\beta(a,P^{\epsilon,h_{bw}}$.

---

### Official Review · Reviewer_jsr3 · 2023-10-30

**Soundness:** 3 good
**Presentation:** 3 good
**Contribution:** 3 good
**Rating:** 8
**Confidence:** 4

**Summary:**

This work studies two very important settings in causal inference: continuous treatment & unmeasured confounding. This work introduces a new kernel-based DR estimator designed for continuous treatments, and it presents an efficient approach to solving nuisance functions and demonstrates the estimator's effectiveness through synthetic and real-world data.

**Strengths:**

Clear writing & solid theoretical results

**Weaknesses:**

The reviewer is not an expert in deep learning for causal inference but would assume there are working targeting the studies setting: causal effect estimation for continuous treatment under potential missing confounders.

The reviewer understands the page limitation and would recommend a detailed comparison to existing literature in the appendix --- how is this method novel and why this novel kernel modification is necessary to handle the pitfalls in the current literature?

The reviewer would like to raise the score once there are more literature survey on that direction and added numerical experiments: comparing with **SOTA** DL method on **benchmark** datasets in the revision.

A minor comment: will it be better to call section 2 "background" instead of "related works"? And it is not clear to the reviewer why "in this paper" is highlighted in the second paragraph of section 2...

**Questions:**

See weakness.

---

> ### Author Response · Authors · 2023-11-18
> **Response to Reviewer jsr3**
>
> Thank you for the positive assessment and valuable suggestions on our paper. We address your questions below.
>
> **Q1.** Recommending a detailed comparison to the existing literature in the appendix  --- how is this method novel and why this novel kernel modification is necessary to handle the pitfalls in the current literature?
>
> **A.** We have claimed in our **contribution** and the second but last paragraph in the introduction that our method for the first time can estimate the causal effect over continuous variables, even if the unboundedness assumption is violated. To better stand out our uniqueness, we have appended a related work section in **Appx. B** that gives a comprehensive introduction to related works and how our method differs from it.
>
> We use kernel modification because its kernel approximation is consistent (Theorem 4.2); more importantly, it can approximate the influence function according to Theorem 4.5.
>
> **Q2.** More  numerical experiments.
>
> **A.**  We conduct experiments on four additional settings, encompassing three synthetic data with different data-generating mechaninisms, as well as a time-series forecasting setting. Besides, we compare with two additional DL-based baselines *MINMAX* and *NMMR*. Results in Tab. 1 and Tab. 3 (Appx. G) suggest that our methods can outperform others in these settings. For completeness, we attach the table of additional experiments in the Appx. G to the rebuttal console:
>
> | Dataset     |            |  Size | PMMR      | KPV       | DFPV      | MINIMAX   | NMMR-V    | POR       | PKIPW     | PKDR      |
> |-------------|------------|-------|-----------|-----------|-----------|-----------|-----------|-----------|-----------|-----------|
> |   |
> |    "Hu et al.  | Scenario 1 | 1000  | 0.25±0.05 | 0.25±0.05 | 0.59±0.35 | 1.45±1.32 | 0.17±0.09 | 0.16±0.23 | 0.29±0.12 | **0.16±0.22** |
> |     (2023)"        | Scenario 2 | 1000  | 0.16±0.02 | 0.16±0.02 | 0.22±0.17 | 0.88±0.29 | **0.05±0.04** | 0.08±0.07 | 0.15±0.06 | 0.07±0.06 |
> |             | Scenario 3 | 1000  | 0.10±0.02 | 0.10±0.02 | 0.28±0.39 | 0.45±0.29 | 0.21±0.10 | 0.22±0.20 | **0.09±0.03** | 0.21±0.19 |
> | Time series |            | 500   | 0.11±0.04 | 0.13±0.12 | 0.20±0.14 | 0.21±0.09 | 0.18±0.12 | 0.10±0.14 | 0.21±0.05 | **0.09±0.12** |
> |             |            | 1000  | 0.10±0.05 | **0.08±0.07** | 0.16±0.21 | 0.22±0.06 | 0.18±0.10 | 0.12±0.12 | 0.20±0.06 | 0.12±0.10 |

---

> > ### Comment · Reviewer_jsr3 · 2023-11-22
> >
> > The reviewer acknowledged that the concerns are addressed and increased the score accordingly.

---

### Author Response · Authors · 2023-11-18
**Response to all reviewers**

We appreciate the invaluable advice and positive feedback from all reviewers. We have updated our manuscript according to your comments. To summarize,

1. Per the request from Reviewer jsr3, we conduct experiments on 4 additional benchmarks, encompassing a time-series forecasting scenario. Our method can achieve comparable or better results in these settings. Please refer to Appx. G for results.
2. Per the request from Reviewer jsr3, we compare with two DL-based benchmarks, *MINMAX* and *NMMR*. According to Tab. 1 and Tab. 3 (in Appx. G), our methods outperform both of them.
3. Per the request from Reviewer jsrs, we append a related-work section in the Appx. B.
4. In addressing the question from Reviewer sZEk, we have made slight modifications to Lemma 5.1 and introduced a new paragraph to enhance clarity.
5. Per the request from Previewer eqxp, we provide the consistent analysis for our estimator in Theorem E.9, which shows that with high probability, our estimator can converge to the ground-truth causal effect.

---

### Meta-Review · Area_Chair_wVzP · 2023-12-04

**Metareview:**

To solve the problem that the current form of doubly robust (DR) estimator is restricted to binary treatments, this paper proposes a kernel-based DR estimator that can handle continuous treatments for proximal causal learning.

A reasonable amount of discussions took place between the authors and the reviewers. In the end, we got four reviews with ratings of 8, 6, and 6 with confidence of 4, 4, and 3 respectively. The reviewers appreciate the novel framework and the comprehensive experiments.

Reviewers proposed issues about experimental results (jsr3), theoretical correctness (sZEk), and motivation (eqxp). Fortunately, the authors have addressed the main issues proposed by the reviewers. The decision is acceptance.

**Justification For Why Not Higher Score:**

The paper is considered good in a specialized group.

**Justification For Why Not Lower Score:**

Most concerns that were raised have been well-addressed by the authors and the reviewers are consensus to be accepted with score at least 6.

---

### Decision · Program_Chairs · 2024-01-16

Accept (poster)